# Automatic Transcription of Handwritten Old Occitan Language

**Esteban Garces Arias**[1]♠  **Vallari Pai**[1]◇  **Matthias Schöffel**[2]♣

**Christian Heumann**[1]♠  **Matthias Aßenmacher**[1,3]♠

[1] Department of Statistics, LMU, Munich, Germany
[2] Bavarian Academy of Sciences, BAdW, Munich, Germany
[3] Munich Center for Machine Learning (MCML), LMU, Munich, Germany

♠`{esteban.garcesarias,chris,matthias}@stat.uni-muenchen.de`
♣`matthias.schoeffel@badw.de`  ◇`Valari.Pai@campus.lmu.de`

## Abstract

While existing neural network-based approaches have shown promising results in Handwritten Text Recognition (HTR) for high-resource languages and standardized/machine-written text, their application to low-resource languages often presents challenges, resulting in reduced effectiveness. In this paper, we propose an innovative HTR approach that leverages the Transformer architecture for recognizing handwritten Old Occitan language. Given the limited availability of data, which comprises only word pairs of graphical variants and lemmas, we develop and rely on elaborate data augmentation techniques for both text and image data. Our model combines a custom-trained Swin image encoder with a BERT text decoder, which we pre-train using a large-scale augmented synthetic data set and fine-tune on the small human-labeled data set. Experimental results reveal that our approach surpasses the performance of current state-of-the-art models for Old Occitan HTR, including open-source Transformer-based models such as a fine-tuned TrOCR and commercial applications like Google Cloud Vision. To nurture further research and development, we make our models, data sets, and code publicly available: https://huggingface.co/misoda

## 1 Introduction

Old Occitan, also known as Old Provençal, was a language widely spoken in the 11th-16th centuries, in particular in southern France, northeastern Spain, and northwestern Italy. It occupies a prominent position in both the linguistic and cultural legacy of Romance languages, primarily due to its role as a precursor to French lyric and the dissemination of works by Troubadours throughout Europe. Despite its well-established historical importance, the linguistic research of Old Occitan remains relatively limited. In contrast to Old French, it lacks comprehensive collections of digitized manuscripts with scanned images and annotated corpora, which are essential resources for conducting detailed morpho-syntactic or syntactic analyses (Scrivner and Kübler, 2012).

More recently, however, there have been efforts to ramp up the availability of resources for this language. An example of this is the creation of a digital version of the Old Occitan dictionary[1], led by a team of researchers at the Bavarian Academy of Sciences. The project aims to establish an open-access database for scientific research, featuring a curated collection of vocabulary. A crucial step for this is the digitization of handwritten material of (non-standardized) words. For our research, we had access to a collection of 600,000 handwritten cards (93% of these being unlabeled) containing graphical variants alongside their respective lemmas. In this work, we explore and combine Transformer-based architectures with data augmentation techniques specifically tailored to address the challenges posed by HTR for low-resource languages. The experimental outcomes demonstrate the effectiveness of our method, as it achieves state-of-the-art (SOTA) results in Old Occitan.

**Contributions**  This work provides the following contributions:

1. We propose a Transformer-based HTR approach using an encoder-decoder architecture, effectively addressing limitations in low-resource languages through data augmentation techniques.

2. We conduct a comparative analysis of various architectures, data augmentation methods, and decoding strategies.

---

[1]*Dictionnaire de l'occitan médiéval*
`http://www.dom-en-ligne.de/`

3. We extensively review existing open-source and closed-source OCR and HTR tools, and benchmark our model against them. Our model achieves state-of-the-art (SOTA) results on the Old Occitan data set.

4. We publish our codebase and the data sets: https://huggingface.co/misoda

## 2 Related Work

A comparative study by Michael et al. (2019) examined various attention mechanisms and positional encodings to address the alignment between input and output sequences in the context of HTR, a field of study that was traditionally dominated by Recurrent Neural Network (RNN) encoders combined with Connectionist Temporal Classification (CTC) decoders (Bluche and Messina, 2017; Graves et al., 2006; Pham et al., 2014).

Subsequently, a systematic literature review conducted by Memon et al. (2020) examined HTR and OCR research between 2000 and 2018. Most of the reported approaches employed architectures based on Convolutional Recurrent Neural Network (CRNN), Long Short-Term Memory (LSTM) models, and CTC, applied to well-studied languages such as English, Chinese, Urdu, and Arabic.

With the popularization of the Transformer model by Vaswani et al. (2017) and its multimodal applications, other approaches deviating from the conventional Convolutional Neural Network (CNN) and RNN architectures have emerged and shown SOTA performances in HTR. One notable example is Transformer OCR (TrOCR) (Li et al., 2021), which adopts a Transformer model as its backbone. This end-to-end HTR model consists of a vision encoder and an autoregressive text decoder. Subsequently, Barrere et al. (2022) proposed a more compact Transformer-based architecture that employs different visual feature embedding techniques and combines CTC and cross-entropy loss during training. Both Transformer-based approaches achieved competitive results on the widely-used IAM data set (Marti and Bunke, 2002), which contains overall 115,320 handwritten English words on 13,353 images. Additionally, Diaz et al. (2021) compared various encoder-decoder models and found that using self-attention in the encoder and a CTC-trained decoder enriched with a language model yielded SOTA performance on the same data set.

Furthermore, researchers have explored the application of Transformer-based models to low-resource languages. For instance, Ströbel et al. (2022) successfully fine-tuned a TrOCR instance for handwritten Medieval Latin and surpassed the widely-used Transkribus (Kahle et al., 2017), a commercial platform for historical document transcription. The results highlight the potential of custom Transformer-based models to address HTR challenges in languages with a limited amount of available data.

## 3 Data Preparation

### 3.1 Handwritten cards

Researchers have continuously compiled the Old Occitan card database used in this study since around 1960. Each card follows a prototypical structure, as exemplified in Figure 1, consisting of 'GRAPHICAL VARIANT → LEMMA' written in uppercase letters against a blue background. The handwritten text, on average, spans 15.7 characters and typically comprises two words: the graphical variant (left side) with an average length of 7.5 characters, and the lemma (the standardized entry in the Old Occitan dictionary; right side) with an average length of 7.2 characters. The two words are always separated by a one-character arrow sign, denoting the reference from the graphical variant to the lemma.

The labeled data consist of 41,634 samples containing 39,554 unique graphical variants and 15,852 unique lemmas. This results in an average of 2.5 graphical variants per lemma. The annotations are provided in plain text format and consist of a set comprising 73 distinct characters. These characters primarily include the 26 letters of the Latin alphabet, diacritics from the French alphabet, and digits. Additionally, the labeled material contains punctuation symbols, question and exclamation marks, parentheses, brackets, and phonetic notation. A comprehensive overview of the character inventory, including their absolute and relative frequencies, can be found in Table 9 in Appendix C.

The majority of the images in the data set hold text that is exclusively written on the upper half of the card, presented in a clear and legible manner. However, a notable portion of the cards exhibits text spread across multiple lines, accompanied by corrections, additional side notes, bent corners, or other irregular marks that may have arisen during

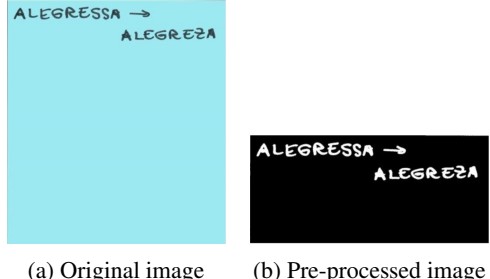

(a) Original image     (b) Pre-processed image

Figure 2: Pre-processing of an original Old Occitan card, after cropping, enhancement of contrast, sharpness and brightness, and binarization.

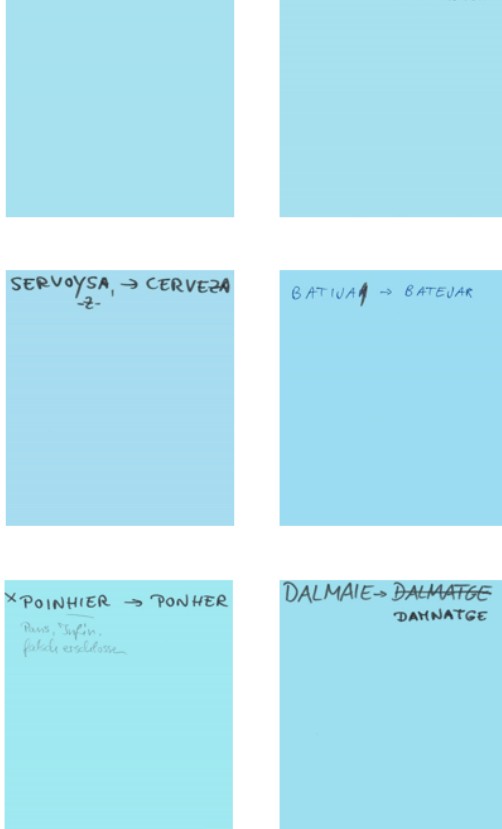

Figure 1: Examples of (original) cards with Old Occitan text. Standard cases are depicted in the first row, while different exceptions are displayed in the images below.

the scanning process. Various examples illustrating these variations are depicted in the lower part of Figure 1. The average dimensions (width × height) of a card are $1235 \times 1390$ pixels, with a print resolution of 300 dpi.

## 3.2 Data pre-processing

We pre-process the data set to enhance model performance, enabling data augmentation and synthetic image generation.

**Image pre-processing** Firstly, the images undergo a cropping process where only the upper 40% is retained for the text recognition task, considering that the relevant text is always located in the upper section of the card, and the lower 60% of the image might contain noise in the form of irregular annotations. Secondly, contrast, sharpness, and brightness enhancements are applied to improve the legibility of the text. Lastly, the images are binarized. To ensure compatibility for data augmentation, we align this binarized variant with the

format used in the EMNIST data set (Cohen et al., 2017), which features white text on a black background. An example of our image pre-processing is shown in Figure 2. A comprehensive list of the pre-processing parameters is shown in Table 6 in Appendix B.

**Corpus pre-processing** First, we create a compilation of different relevant corpora: We combine excerpts from *Histoire de la langue provençale: à Avignon du XIIe au XIXe siècle* (Pansier, 1974) with three Old Occitan corpora publicly available[2]. The corpora are composed of documents covering literature, lyric, law, ecclesiastic narrative, and administration texts from 1050-1550. A detailed overview of the corpora used (name, genre, and number of tokens) is shown in Table 11 in Appendix E. Second, we apply additional pre-processing steps to ensure the cleanliness and continuity of the text. Misplaced empty spaces, such as those found after apostrophes or between words and punctuation marks, are removed. Additionally, we eliminate line breaks to create uninterrupted text. Finally, the set of unique words is extracted and transformed into an upper-case format. These words are then combined with every other word to generate synthetic word pairs, separated by an arrow symbol to emulate the structure of 'GRAPHICAL VARIANT → LEMMA'. This procedure results in the creation of over 82 million synthetic word pairs, from which synthetic labels were generated for data augmentation purposes. Table 12 in Appendix E provides an example of this process. It is worth noting that initially, our approach involved the generation of word pairs with a maximum Levenshtein distance of two, as this distance closely aligned with the median value observed in our training labels. However, we subsequently decided to remove

---

[2] http://cl.indiana.edu:8080/annis-gui-3.6.0/

this constraint to increase not only the quantity and variability of word pairs but also the potential generalization of our model, in particular when applied to slightly different data. A noteworthy result of this modification can be observed in the case of special characters such as Ä, Ö, and French diacritics, which experienced a significant increase in examples, growing from just a few dozen to thousands.

### 3.3 Data Augmentation

**Real images**   We consider two steps to augment the real pre-processed images: random rotation and dilation. The parameters are depicted in Table 10 while selected examples of augmented images are displayed in Figure 5, both in Appendix D.

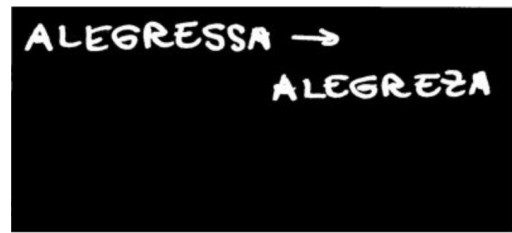

(a) Real pre-processed image

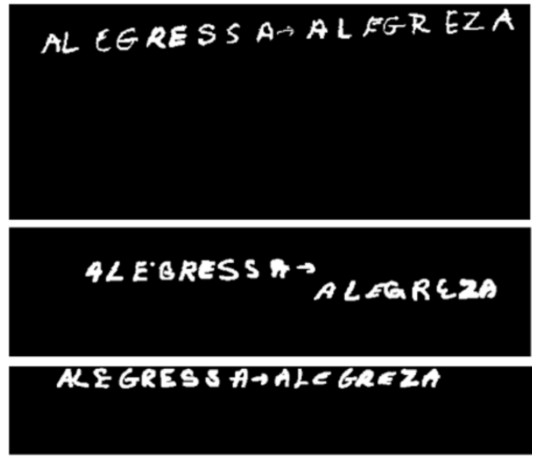

(b) Synthetic images generated with EMNIST

Figure 3: Examples of pre-processed original images and synthetic images generated from the EMNIST data set and the training labels. Synthetic images were also generated from a synthetic corpus (cf. Sec. 3.2).

**Synthetic images**   To increase the available training data, we expand the EMNIST data set by incorporating special characters frequently encountered in our cards, including the arrow (→) and the cedilla (Ç). This data set, the training labels, and the pre-processed corpus described in Section 3.2, are used to generate synthetic images.

The generation process involves utilizing real training labels (or synthetic labels from the augmented corpus) as input. For each character in these labels, a random image is sampled from the extended EMNIST character inventory. Concatenating these images produces the synthetic output image. To replicate the variability observed in real Old Occitan handwritten text, random components are introduced, such as varying image sizes, number of lines, rotation angles, dilation factors, and pixel distances between characters, allowing for overlapping and irregular spacing. This approach enables the generation of an average of 37.5 images per second. An example of the generated images is depicted in Figure 3.

For reproducibility purposes, a comprehensive overview of the image generation parameters is provided in Table 13, while the steps are outlined in Algorithm 1. The code for image generation, along with the extended version of EMNIST, is publicly accessible[3].

**Final data set**   Around 7% of the handwritten material was labeled, amounting to 41,634 samples. Among these, we use 80% (33,308) for training, 10% (4,163) for validation, and the remaining 10% (4,163) for testing. To expand the training data, we generate synthetic images. Specifically, we generate 180,000 images during the model selection phase. For enhanced pre-training of the best-performing model, we generate additional 720,000 images, resulting in a total of 900,000 images.

**Stage 1: Model selection**   This stage involves exploring various combinations of vision models as encoders and language models as decoders. The pre-training, fine-tuning and evaluation configurations can be found in Table 1.

**Stage 2: Final model training**   Once the optimal encoder-decoder combination was identified, we enhanced the data augmentation for (a) pre-training, with more synthetic images, and (b) fine-tuning, incorporating random rotation and dilation of real images, as described in Table 1.

## 4   Experimental Setup

**Model selection**   We explore combinations of four vision encoder models: BEiT (Bao et al., 2021), DeiT (Touvron et al., 2021), ViT (Dosovitskiy et al., 2021), and Swin (Liu et al., 2021)

---

[3]Public GitHub repository

| | Stage 1
(Model selection) | Stage 2
(Final model training) |
|---|---|---|
| Pre-training
(w/ synthetic images) | 180,000 | 900,000 |
| Fine-tuning
(w/ real images) | 33,308 | $33,308 \times 7$
(w/ random rotations
and dilations, cf. Table 10) |
| Evaluation
(w/ real images) | 4,163
(Validation) | 4,163
(Test) |
| Total | 217,471 | 1,137,319 |

Table 1: Data configuration for the two stages of this research: Model selection and final model training.

with two language decoders: GPT-2 (Radford et al., 2019) and BERT (Devlin et al., 2019). To investigate the impact of synthetic data, we train each combination using three different training setups: synthetic data only ("*pre-train*"), real data only ("*only real*"), and fine-tuning the pre-trained model with real data ("*fine-tune*"). This resulted in a total of $4 \times 2 \times 3 = 24$ experiments.

To evaluate the performance of the models, we used our validation data set. Tables 5 and 7 provide a comprehensive overview of the model combinations and the corresponding training regimes.

**Final model training** After selecting the best-performing model on the validation set, we use enhanced data augmentation and incorporate 900,000 synthetic images in total during pre-training. Seven random rotations and dilations were applied to the real images, as outlined in Table 1.

To assess the effectiveness of our optimized model, we conducted a benchmarking study. We compared its performance on the test set against popular open-source tools that support handwritten Occitan. In addition, we included a commercial alternative, Google Cloud Vision[4], known for its proven high performance in practical applications (Thammarak et al., 2022). A short description of the models is presented in Table 2.

| Model | Open-Source | Architecture | Fine-tuned w/ our data |
|---|---|---|---|
| EasyOCR (Jaided, 2020) | Yes | ResNet+CTC | No |
| Tesseract OCR (Ooms, 2023) | Yes | CNN+LSTMs | No |
| PaddleOCR (Du et al., 2020) | Yes | CRNN+CTC | No |
| R18+LSTM (He et al., 2016) | Yes | ResNet18+CNN+LSTM | Yes |
| TrOCR (Li et al., 2021) | Yes | Transformer | Yes |
| Google Cloud Vision | No | Unknown
(commercial) | No |

Table 2: Characteristics of models used for benchmarking, featuring open-source and commercial tools.

---

[4]https://cloud.google.com/vision/docs/handwriting

## 4.1 Model Architectures

**Tokenizer** We employ byte-level BPE (Byte Pair Encoding) tokenization (Sennrich et al., 2016) to train a tokenizer using our training labels. This technique involves iteratively merging the most frequent pairs of bytes until a predetermined limit is reached, or no further merges are possible. In our case, we set the number of output tokens to 73 (cf. Table 9) to match the number of characters observed in our labels (note that in this non-standardized, low-resource language, the vocabulary of words is not entirely known). The tokenizer required 82 encodings for classification, padding, end-of-sentence, and unknown tokens, as well as for the observed characters, given that some of them require more than a single integer for encoding (cf. Table 14 in Appendix G).

**Image encoder** We use the Swin Transformer (27.5M parameters), an extension of ViT (86.4M parameters), to enhance efficiency and enable cross-window connections. Swin incorporates hierarchical feature maps and shifted window attention, limiting self-attention to non-overlapping local windows. Like ViT, BEiT, and DeiT, the image is divided into patches and projected into embeddings. However, Swin uses smaller patches to avoid high computational costs. Each patch is assigned to a self-attention window for local processing, reducing the complexity of the self-attention. To capture features across windows, the self-attention window is shifted, allowing for the processing of separated image regions. This hierarchical structure enables lower blocks to handle fine-grained information while upper layers operate on merged visual representations. Our model uses a newly initialized Swin encoder with a pre-trained image processor (*swin-base-patch4-window7-224-in22k*).

**Text decoder** We utilize a BERT-based decoder architecture (114.5M parameters) for our text decoding task. This decoder employs an autoregressive language modeling objective, and the probabilities over the vocabulary are calculated using the softmax function. During decoding, we do not apply n-gram repetition penalties. This choice is motivated by the presence of repetitive patterns in our handwritten material, as indicated by the median Levenshtein distance (Levenshtein et al., 1966) of two between graphical variants and the lemmas. To effectively capture the textual structure of the task, we choose to train our decoder from scratch.

| Encoder | Decoder | Training Setup | GPU Utilization (%) | Training Runtime (h) | Inference Time (s) | # Examples / s | CER (Weighted) | Correctly predicted labels (%) |
|---|---|---|---|---|---|---|---|---|
| BeiT | GPT-2 | Pre-Train | 31.9 | 19.9 | 1,718.9 | 2.52 | 0.633 | 0.2% |
| | | Only Real | 31.4 | 14.8 | 1,718.9 | 2.42 | 0.406 | 6.1% |
| | | Fine-Tune | 32.2 | 5.8 | 1,651.1 | 2.52 | 0.215 | 21.6% |
| | BERT | Pre-Train | 38.7 | 18.2 | 1,356.3 | 3.07 | 0.676 | 0.1% |
| | | Only Real | 29.5 | 14.4 | 1,374.8 | 3.03 | 0.455 | 4.1% |
| | | Fine-Tune | 28.5 | 6.23 | 1,360.8 | **3.06** | 0.273 | 15.5% |
| DeiT | GPT-2 | Pre-Train | 34.6 | 20.1 | 1,599.5 | 2.60 | **0.279** | **14.1%** |
| | | Only Real | 27.8 | 18.6 | 1,683.9 | 2.47 | 0.041 | 71.2% |
| | | Fine-Tune | 29.8 | 6.2 | 1,698.8 | 2.45 | 0.021 | 82.8% |
| | BERT | Pre-Train | 38.1 | 20.5 | **1,301.3** | **3.20** | 0.290 | 12.9% |
| | | Only Real | 16.5 | 16.5 | **1,361.7** | **3.06** | 0.041 | 71.4% |
| | | Fine-Tune | 27.4 | 6.3 | 1,370.1 | 3.04 | 0.021 | 81.9% |
| ViT | GPT-2 | Pre-Train | 41.5 | 19.0 | 1,619.3 | 2.57 | 0.368 | 6.0% |
| | | Only Real | 30.0 | 15.8 | 1,732.3 | 2.40 | 0.029 | 79.4% |
| | | Fine-Tune | 30.7 | 6.1 | 1,655.7 | 2.51 | 0.022 | 83.7% |
| | BERT | Pre-Train | 36.6 | 19.2 | 1,305.1 | 3.19 | 0.338 | 7.2% |
| | | Only Real | 26.0 | 18.6 | 1,364.9 | 3.05 | **0.023** | **83.4%** |
| | | Fine-Tune | 28.9 | 5.9 | 1,371.1 | 3.04 | 0.018 | 86.3% |
| Swin | GPT-2 | Pre-Train | 30.9 | 20.0 | 1,601.3 | 2.60 | 0.336 | 9.0% |
| | | Only Real | 22.2 | 19.9 | 1,744.8 | 2.39 | 0.026 | 80.6% |
| | | Fine-Tune | 24.0 | 6.0 | 1,661.8 | 2.50 | 0.020 | 84.2% |
| | **BERT** | Pre-Train | **25.5** | **24.0** | 1,353.0 | 3.08 | 0.339 | 8.0% |
| | | Only Real | **21.2** | **21.0** | 1,407.7 | 2.96 | 0.026 | 81.1% |
| | | Fine-Tune | **22.9** | **6.5** | 1,400.3 | 2.97 | **0.016** | **87.0%** |

Table 3: Results for all model combinations trained on synthetic and/or real data. The best results are highlighted in **bold**. All experiments were performed with a GPU NVIDIA Tesla V100 (16 GB).

This approach enables us to prioritize the unique characteristics of the text and utilize a customized tokenizer specifically trained to excel at a granular character level. Additionally, we explore the use of GPT-2 (114.3M parameters) as an alternative to the BERT-based decoder.

**Performance metrics**  We assess the models using multiple metrics, including weighted Character Error Rate (CER) and the percentage of correctly predicted labels. Additionally, we measure training runtime, GPU utilization (%), total number of trainable parameters, and inference speed in labels per second to gauge resource usage and complexity.

The CER is computed by summing up edit operations and dividing by the label length.

$$CER = \frac{S + D + I}{N} = \frac{S + D + I}{S + D + C},\quad (1)$$

where $S$ is the number of substitutions, $D$ is the number of deletions, $I$ is number of insertions, $C$ is the number of correct characters, and $N$ is number of characters in the label. To account for the varying length of the labels, which range from 4 to 40 characters with a standard deviation of 4.27, we utilize the weighted CER.

$$WeightedCER = \frac{\sum_{i=1}^{n} l_i * CER_i}{\sum_{i=1}^{n} l_i},\quad (2)$$

where $l_i =$ is the number of characters of label $i$, and $CER_i$ is the CER for example $i, i = 1, \ldots, n$. The observed values for all defined metrics are summarized in Tables 3 and 5. For reproducibility, Table 7 contains a list of hyperparameter values used during training.

**Decoding strategy**  We considered a beam search strategy and experimented with different beam width values: 1 (greedy), 4, 10, 15, 30, and 50. The additional parameters for natural language generation are summarized in Table 8 in Appendix B.

## 5 Results

### 5.1 Model selection

The results of the comparative experiments are presented in Table 3, demonstrating that the combination of Swin and BERT yields the best outcomes across various evaluation metrics. These metrics include weighted CER (0.016), percentage of correctly predicted labels (87.0%), lowest runtime (6.5 hours), GPU utilization (22.9%), and the number of parameters (142M, second only to Swin + GPT-2). Regarding the inference speed, the model predicts 2.97 labels/second on average, not far away from the fastest model (BEiT + BERT, with 3.06 predicted labels/second). The results also highlight the effectiveness of using synthetic data, as it decreases the weighted CER from 0.026 ("*only real*") to 0.016 ("*fine-tune*") and increases the percentage of correctly predicted labels from 81.1% ("*only real*") to 87.0% ("*fine-tune*").[5]

---

[5] Similar trends of improvements when using synthetic data are observed for the majority of model combinations.

## 5.2 Benchmarking against HTR & OCR tools

We further enhance data augmentation on the Swin + BERT model, as outlined in Table 1. Subsequently, we compare the performance of our model with open-source and closed-source alternatives. The benchmarking results, presented in Figure 4, demonstrate the superior performance of our model with respect to the competing tools.

Our approach achieves better results across all key evaluation metrics, including CER (0.004), weighted CER (0.005), and the percentage of correctly predicted labels (96.5%). Additionally, our model exhibits enhanced stability and robustness, with the lowest standard deviation (0.029) among all competing models. Remarkably, our model outperforms fine-tuned Transformer-based alternatives like TrOCR and fine-tuned conventional alternatives like R18+LSTM. Additionally, it surpasses the performance of widely-adopted OCR and HTR tools designed to handle handwritten Occitan.

Note that the fine-tuned TrOCR, despite having a relatively high weighted CER (0.083), achieves a 94% share of correctly predicted labels. However, the remaining 6% (incorrect) predictions exhibit remarkably high CER values, shown by the large standard deviation. We attribute this to the fact that it was pre-trained with millions of printed and handwritten material, which included sequences that were considerably longer than those found in the smaller Old Occitan dataset. On the other hand, the more conventional fine-tuned R18+LSTM model shows relatively low weighted CER values (0.043) but a rather constrained proportion of accurately predicted labels (57.2%), especially when compared to the Transformer-based alternatives.

## 5.3 Error analysis

To comprehensively assess the strengths and weaknesses of our model, we conducted an error analysis across various dimensions, including the number of handwritten lines, annotation quality, decoding strategy, label length, and character representation in the labels (cf. Figures in Appendix H for visualization). Moreover, we highlight the ten worst performances and the top ten performances on complex images (Figures 12 and 13 in Appendix H). Overall, the results suggest a negative correlation between the number of text lines and the model performance. When processing a single line of text, the model achieved a mean weighted CER of 0.003 and accurately predicted 97.1% of the labels. However, as the number of text lines increased up to three, the model's performance declined, resulting in a mean weighted CER of 0.161 and a correct prediction rate of 71.4%.

Additionally, the quality of annotations proved to be an important factor affecting performance. For standard images, the model achieved an average weighted CER of 0.004 and accurately predicted 97.0% of the labels. However, when dealing with images containing annotation errors or noise, the performance decreased to a mean weighted CER of 0.043, with a correct prediction rate of 81.2%. It is important to note that these observations were based on a limited number of examples, with irregular annotation quality and three lines of text comprising 14 and 138 instances, respectively. Additionally, it should be mentioned that a median CER value of zero was observed in images featuring three lines of text and in those with annotation irregularities. This suggests that although these particular image types present a challenge to the model, it exhibits a proficient capability in handling the majority of such exceptions.

Further, we observe that in cases where predictions were correct, the label length averaged 15.7 characters, while incorrect predictions had an average label length of 18.3 characters. Additionally, since the impact of beam width on the decoding quality was low, we decided to use greedy search as a decoding strategy for efficiency. Finally, we observed that unsurprisingly, higher weighted CER values were associated with less frequently appearing characters.

## 5.4 Ablation study

In order to assess the influence of data augmentation and synthetic data independent of each other, we use the Swin + BERT model trained solely on real images as a baseline for our ablations. We systematically introduce the individual steps and finally combine them for our final model (cf. Tab. 4). The number of synthetic images appears to be the most influential factor, highlighted in bold. Specifically, incorporating 900,000 synthetic examples resulted in a reduction of the weighted CER by 0.02 and in an increase of correctly predicted labels by 13.2 percentage points. The impact of random rotation and dilation on real images was found to be comparably low, improving the weighted CER by 0.002 and the share of correctly predicted labels by 2.4 percentage points. Note that the effect of

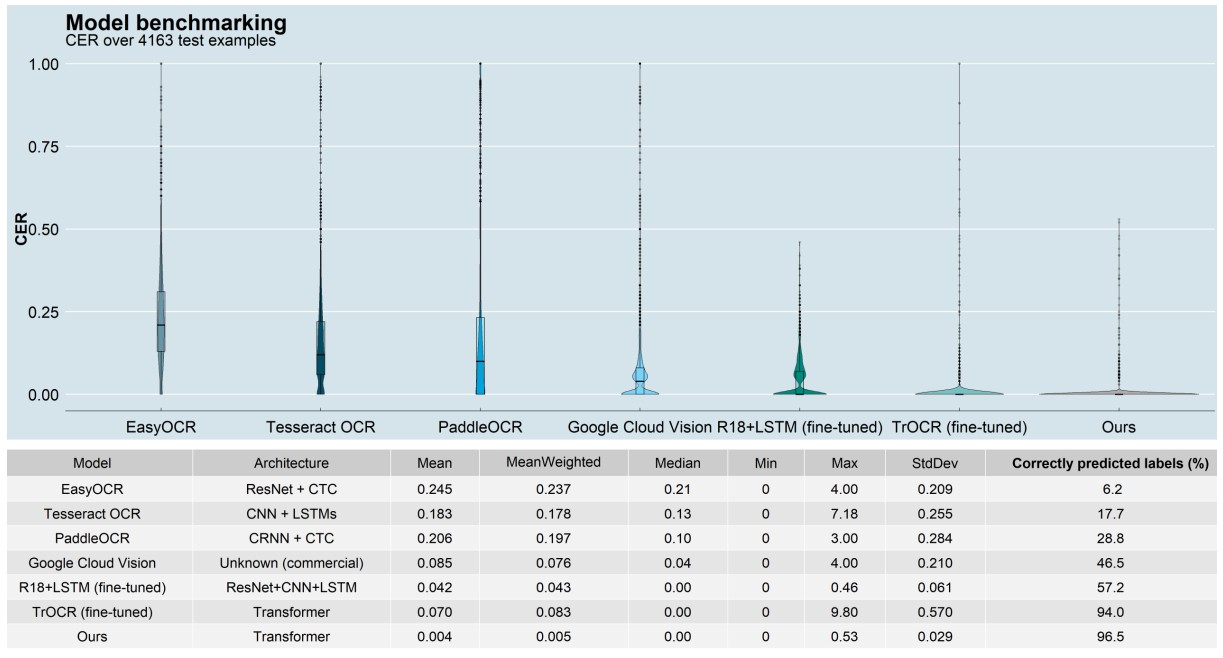

| Model | Architecture | Mean | MeanWeighted | Median | Min | Max | StdDev | Correctly predicted labels (%) |
|---|---|---|---|---|---|---|---|---|
| EasyOCR | ResNet + CTC | 0.245 | 0.237 | 0.21 | 0 | 4.00 | 0.209 | 6.2 |
| Tesseract OCR | CNN + LSTMs | 0.183 | 0.178 | 0.13 | 0 | 7.18 | 0.255 | 17.7 |
| PaddleOCR | CRNN + CTC | 0.206 | 0.197 | 0.10 | 0 | 3.00 | 0.284 | 28.8 |
| Google Cloud Vision | Unknown (commercial) | 0.085 | 0.076 | 0.04 | 0 | 4.00 | 0.210 | 46.5 |
| R18+LSTM (fine-tuned) | ResNet+CNN+LSTM | 0.042 | 0.043 | 0.00 | 0 | 0.46 | 0.061 | 57.2 |
| TrOCR (fine-tuned) | Transformer | 0.070 | 0.083 | 0.00 | 0 | 9.80 | 0.570 | 94.0 |
| Ours | Transformer | 0.004 | 0.005 | 0.00 | 0 | 0.53 | 0.029 | 96.5 |

Figure 4: Comparison of test set performance between our model and other OCR and HTR tools supporting handwritten Occitan language. To aid visualization, the y-axis is limited to values between 0.0 and 1.0. Thus it is worth noting that certain models may have CER values greater than 1.0 (which are cut off here).

the initial 180,000 images accounts for half of the increase observed for the 900,000 synthetic images.

| Architecture: *Swin + BERT* | CER (Weighted) | Correctly predicted (%) |
|---|---|---|
| Only Real | 0.026 | 81.1% |
| Only Real (w/ augmentation) | 0.024 | 83.5% |
| w/ Synthetic (180,000) | 0.016 | 87.0% |
| **w/ Synthetic (900,000)** | **0.006** | **94.3%** |
| Final model | 0.005 | 96.5% |

Table 4: Impact of (amount of) synthetic data and augmentation (most beneficial step highlighted in **bold**).

### 5.5 Inspecting domain shift

To assess the generalization capabilities of our model, we conduct tests on a different Old Occitan data set (cf. Fig. 14 in Appendix I). This data set consists of 316 images containing upper-case single lemmas with a mean length of 7.6 characters and without any graphical variants or arrows. The text is written on green, yellow, or red backgrounds. The predictions of our model undergo automatic post-processing to remove any predicted arrows or any predicted text after an arrow. The results exemplify good generalization of our model in this relatively similar data set, with weighted CER values ranging from 0.010 to 0.011 and a percentage of correctly predicted labels ranging

from 91.6% to 92.3% (cf. Fig. 15 in Appendix I). This data set can also be accessed publicly https://huggingface.co/misoda

## 6 Discussion and Outlook

HTR encounters substantial challenges in low-resource languages, hindering language diversity and inclusivity. These challenges arise from diverse handwriting styles, variable image quality, the absence of a standardized vocabulary, and limited training data. In response, we propose a customized end-to-end tool specifically designed for the Old Occitan dictionary. However, our approach can also serve as a blueprint for similar systems for other (extremely) low-resource languages.

To further enhance the performance and capabilities of our model, there are several key areas that can be addressed. Firstly, investigating penalization techniques in the context of low-resource languages is a promising path to prevent overfitting, improve model generalization, and reduce potential biases in the predictions (Steiner et al., 2021). Secondly, considering smaller architectures, such as DistilBERT (Sanh et al., 2019), is a strategy worth exploring. Our results have shown that relatively small models achieve the best performances on the Old Occitan data set, indicating that these architectures can offer improved efficiency with-

out compromising performance. Additionally, incorporating SOTA segmentation techniques, such as YOLOv8 (Jocher et al., 2023), SAM (Kirillov et al., 2023), and SEEM (Zou et al., 2023), can potentially bolster the encoding efficiency of the model. This strategy may prove beneficial in scenarios where data availability is constrained, aiding the training process and potentially enhancing performance. Finally, we would like to explore further data augmentation techniques, such as GAN-based approaches, as described by Guan et al. (2020), or grid-based distortion augmentation, as proposed by Wigington et al. (2017), in combination with our own method.

## 7   Conclusion

In this paper, we present a Transformer-based model for the HTR of Old Occitan, a low-resource language. Our approach involves training various configurations of vision encoders in conjunction with language decoders. Through our experiments, we have determined that the most effective combination is a Swin + BERT model, as it achieves superior performance in terms of weighted CER and share of correctly predicted labels. Furthermore, it has fewer parameters, requires less training time, and utilizes fewer computational resources. To improve the performance of our model, we applied elaborate data augmentation techniques, reducing the weighted CER from 0.026 to 0.005 and improving the share of correctly predicted labels from 81.1% to 96.5%. As a result, our approach surpasses various open-source and commercial HTR and OCR tools. During the evaluation of our best model, we observe that it tends to be less effective on images that contain long character sequences, annotation irregularities, multiple lines of text, and rare characters. However, it demonstrates competency in handling complex cases (cf. Fig. 13 in Appendix H). Overall, our approach presents a promising HTR solution for Old Occitan, and nurtures further research to explore applications to other languages and data sets.

## Limitations

Our approach has several limitations that can be addressed to improve its efficiency further. Firstly, the character recognition performance shows high variability, especially for less frequently observed classes, such as the equal sign, brackets, apostrophes, and French diacritics. The model struggles to consistently recognize these cases accurately. To address the model's weaknesses, we suggest further data augmentation to enhance its performance in underrepresented classes.

Secondly, the model generalization potential seems limited due to its training on a specific data structure. The training data primarily consists of uppercase text and predominantly includes word pairs with an arrow in between. This specialization may hinder the ability to generalize and accurately process other types of text and language structures. However, it is worth noting that the model has shown good performance (after post-processing) on a different Old Occitan data set, as demonstrated in Figure 15.

Furthermore, it is important to acknowledge that, except for TrOCR and R18+LSTM, the tools used for comparison in this study were not specifically fine-tuned on the same data set as our proposed Transformer-based approach. This distinction raises questions about how the performance of these tools would compare when applied to the same data. The benchmarking conducted in this study includes tools specialized in OCR that also support HTR. A more comprehensive comparison would involve tools specialized in HTR and fine-tuning them on our own data. Finally, it is important to note that our approach has not been explored for other low-resource languages. Therefore, its performance, when applied to languages beyond the scope of our study, remains uncertain. However, other experiments have shown the good performance of similar Transformer-based models, such as their application on Medieval Latin (Ströbel et al., 2022).

## Ethics Statement

We affirm that our research adheres to the ACL Ethics Policy. This work involves the use of publicly available datasets and does not involve human subjects or any personally identifiable information. We declare that we have no conflicts of interest that could potentially influence the outcomes, interpretations, or conclusions of this research. All funding sources supporting this study are acknowledged in the acknowledgments section. We have made our best effort to document our methodology, experiments, and results accurately and are committed to sharing our code, data, and other relevant resources to foster reproducibility and further advancements in research.

## Acknowledgements

We wish to extend our thanks to the Bavarian Academy of Sciences and the ALMA project for their support and granting us access to the handwritten material. We would like to express our appreciation to Paula Ruppert for her invaluable assistance in preparing and curating the dataset, and to Meimingwei Li for his essential technical support while evaluating CNN+LSTM-based approaches. This work has received partial funding from the Deutsche Forschungsgemeinschaft (DFG, German Research Foundation) as part of BERD@NFDI, under grant number 460037581.

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

# Appendix

## A  Experimental setup

| Encoder | Decoder | # Parameters Encoder + Decoder | Training Setup | # Examples | # Epochs |
|---|---|---|---|---|---|
| BEiT | GPT-2 | 85.7M + 114.3M | Pre-Train | 180,000 | 25 |
| | | | Only Real | 33,308 | 50 |
| | | | Fine-Tune | 213,308 | 50 |
| | BERT | 85.7M + 114.5M | Pre-Train | 180,000 | 25 |
| | | | Only Real | 33,308 | 50 |
| | | | Fine-Tune | 213,308 | 50 |
| DeiT | GPT-2 | 86.4M + 114.3M | Pre-Train | 180,000 | 25 |
| | | | Only Real | 33,308 | 50 |
| | | | Fine-Tune | 213,308 | 50 |
| | BERT | 86.4M + 114.5M | Pre-Train | 180,000 | 25 |
| | | | Only Real | 33,308 | 50 |
| | | | Fine-Tune | 213,308 | 50 |
| ViT | GPT-2 | 86.4M + 114.3M | Pre-Train | 180,000 | 25 |
| | | | Only Real | 33,308 | 50 |
| | | | Fine-Tune | 213,308 | 50 |
| | BERT | 86.4M + 114.5M | Pre-Train | 180,000 | 25 |
| | | | Only Real | 33,308 | 50 |
| | | | Fine-Tune | 213,308 | 50 |
| Swin | GPT-2 | 27.5M + 114.3M | Pre-Train | 180,000 | 25 |
| | | | Only Real | 33,308 | 50 |
| | | | Fine-Tune | 213,308 | 50 |
| | BERT | 27.5M + 114.5M | Pre-Train | 180,000 | 25 |
| | | | Only Real | 33,308 | 50 |
| | | | Fine-Tune | 213,308 | 50 |

Table 5: Combination of different vision encoder and language decoder models trained on synthetic and/or real data. All the experiments were performed with a GPU NVIDIA Tesla V100 (16 GB).

## B  Hyperparameters

**Parameters for Image Pre-processing**

| Parameter | Value |
|---|---|
| Cropping | image_height*0.40 |
| Contrast Factor | 5 |
| Sharpness Factor | 5 |
| Brightness Factor | 3 |

Table 6: Parameters for image pre-processing. We used the defaults from the PIL (9.2.0) library for all non-reported values.

**Parameters for Training with and without Augmentation**

| Parameter | Value |
|---|---|
| Seed | 42 |
| Optimizer | AdamW |
| Encoder | {BEiT, DeiT, ViT, Swin} |
| Decoder | {GPT-2, BERT} |
| Batch Size (Train, Validation & Test) | 48 |

Table 7: Parameters for training to reproduce our results. For all non-reported values, we used the defaults from the transformers (4.25.1) library.

**Parameters for Natural Language Generation**

| Parameter | Value |
|---|---|
| Max Length | 200 |
| Early Stopping | True |
| No Repeat Ngram Size | 100 |
| Number of Beams | {1, 4, 10, 15, 30, 50} |

Table 8: Parameters for natural language generation. We used the defaults from the `transformers` (4.25.1) library for all non-reported values.

## C   Character Inventory

| # | Character | Frequency | Relative Frequency (%) | # | Character | Frequency | Relative Frequency (%) |
|---|---|---|---|---|---|---|---|
| 1 | A | 88295 | 12.6753 | 38 | ? | 104 | 0.0149 |
| 2 | @ | 83268 | 11.9536 | 39 | , | 103 | 0.0148 |
| 3 | E | 77563 | 11.1346 | 40 | Ü | 77 | 0.0111 |
| 4 | R | 61007 | 8.7579 | 41 | ’ | 56 | 0.0080 |
| 5 | S | 46037 | 6.6089 | 42 | 3 | 45 | 0.0065 |
| 6 | I | 45298 | 6.5028 | 43 | = | 43 | 0.0062 |
| 7 | N | 39317 | 5.6442 | 44 | Ó | 43 | 0.0062 |
| 8 | O | 33621 | 4.8265 | 45 | Ä | 39 | 0.0056 |
| 9 | L | 26316 | 3.7778 | 46 | Ṇ | 30 | 0.0043 |
| 10 | T | 26061 | 3.7412 | 47 | / | 28 | 0.0040 |
| 11 | D | 21790 | 3.1281 | 48 | [ | 19 | 0.0027 |
| 12 | M | 21058 | 3.0230 | 49 | ] | 19 | 0.0027 |
| 13 | C | 21052 | 3.0221 | 50 | È | 18 | 0.0026 |
| 14 | B | 16061 | 2.3057 | 51 | K | 12 | 0.0017 |
| 15 | U | 16004 | 2.2975 | 52 | Ë | 9 | 0.0013 |
| 16 | P | 15598 | 2.2392 | 53 | Ô | 6 | 0.0009 |
| 17 | F | 12039 | 1.7283 | 54 | 4 | 4 | 0.0006 |
| 18 | G | 10935 | 1.5698 | 55 | Â | 4 | 0.0006 |
| 19 | H | 9348 | 1.3420 | 56 | Ò | 4 | 0.0006 |
| 20 | Z | 8583 | 1.2321 | 57 | Ñ | 3 | 0.0004 |
| 21 | V | 4771 | 0.6849 | 58 | Í | 3 | 0.0004 |
| 22 | Y | 3693 | 0.5302 | 59 | Ú | 3 | 0.0004 |
| 23 | J | 2608 | 0.3744 | 60 | ~ | 3 | 0.0004 |
| 24 | Q | 1458 | 0.2093 | 61 | … | 3 | 0.0004 |
| 25 | 1 | 654 | 0.0939 | 62 | + | 3 | 0.0004 |
| 26 | - | 514 | 0.0738 | 63 | 9 | 2 | 0.0003 |
| 27 | X | 477 | 0.0685 | 64 | À | 2 | 0.0003 |
| 28 | 2 | 373 | 0.0535 | 65 | 5 | 1 | 0.0001 |
| 29 | * | 372 | 0.0534 | 66 | Î | 1 | 0.0001 |
| 30 | Ç | 371 | 0.0533 | 67 | " | 1 | 0.0001 |
| 31 | ( | 280 | 0.0402 | 68 | 8 | 1 | 0.0001 |
| 32 | ) | 280 | 0.0402 | 69 | Ÿ | 1 | 0.0001 |
| 33 | Á | 190 | 0.0273 | 70 | 7 | 1 | 0.0001 |
| 34 | Ï | 189 | 0.0271 | 71 | Ö | 1 | 0.0001 |
| 35 |   | 171 | 0.0245 | 72 | < | 1 | 0.0001 |
| 36 | . | 125 | 0.0179 | 73 | W | 1 | 0.0001 |
| 37 | É | 121 | 0.0174 | | | | |

Table 9: Character frequency and relative frequency observed in the labels.

# D   Augmentation of real images

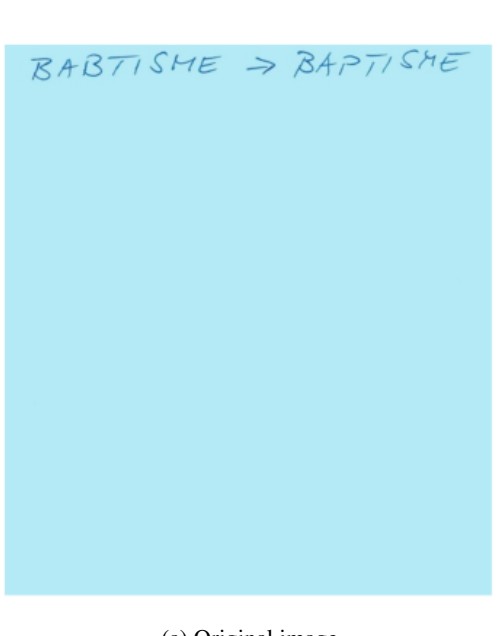

(a) Original image

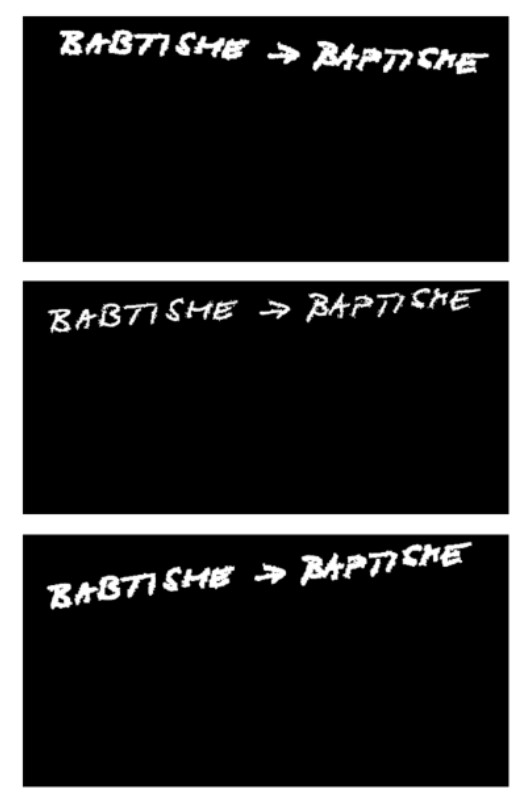

(b) Pre-processed and augmented images

Figure 5: Examples of data augmentation on real images. Random dilations and rotations are applied to the pre-processed images.

| Parameter | Value |
|---|---|
| Seed | 42 |
| Rotation angle | {-4, -3, -2, 0, 2, 3, 4} |
| Dilation factor | {1, 3, 5, 7} |

Table 10: Parameters for data augmentation of real images. We used the defaults from the `PIL` (9.2.0) library for all non-reported values.

# E   Generation of synthetic corpus

| Name | Genre | # Tokens |
|------|-------|----------|
| Lines1-992 | Literary Prose in verse (13th-century) | 6,878 |
| Flamenca (lines 993-2133) | Literary Prose in verse (13th-century) | 8,132 |
| Boece | Verse (10th-century) | 7,054 |
| Pansier | Literary Prose and Administrative Texts (10th to 15th-century) | 49,067 |
| | Total | 71,131 |

Table 11: Compilation of Old Occitan corpora for data augmentation purposes.

| Real corpus (Excerpt) | Synthetic corpus (Excerpt) |
|------------------------|----------------------------|
| Carta audir descebre ne altre kanonegue oi maiso maison sanct adenant nom castel honor aver adenant castel territorio... | CARTA@AUDIR  CARTA@DESCEBRE CARTA@NE  CARTA@ALTRE CARTA@KANONEGUE  CARTA@OI CARTA@MAISO  CARTA@MAISON CARTA@SANCT  CARTA@ADENANT CARTA@NOM  CARTA@CASTEL CARTA@HONOR  CARTA@AVER CARTA@ADENANT  CARTA@CASTEL CARTA@TERRITORIO  AUDIR@CARTA AUDIR@DESCEBRE  AUDIR@NE AUDIR@ALTRE  AUDIR@KANONEGUE AUDIR@OI  AUDIR@MAISO AUDIR@MAISON  AUDIR@SANCT AUDIR@ADENANT  AUDIR@NOM AUDIR@CASTEL  AUDIR@HONOR AUDIR@AVER  AUDIR@ADENANT AUDIR@CASTEL 

 ..... 
 CASTEL@HONOR  CASTEL@AVER CASTEL@ADENANT CASTEL@TERRITORIO TERRITORIO@CARTA TERRITORIO@AUDIR TERRITORIO@DESCEBRE TERRITORIO@NE  TERRITORIO@ALTRE TERRITORIO@KANONEGUE TERRITORIO@OI  TERRITORIO@MAISO TERRITORIO@MAISON TERRITORIO@SANCT TERRITORIO@ADENANT TERRITORIO@NOM TERRITORIO@CASTEL TERRITORIO@HONOR TERRITORIO@AVER TERRITORIO@ADENANT |

Table 12: Example of synthetic corpus generated from a random combination of Old Occitan words in uppercase. The @ sign represents an arrow linking the graphical variant with its lemma.

## F  Generation of synthetic images

| Parameter | Value |
|---|---|
| Seed | 42 |
| Pixel distance (between subsequent characters) | {0, 1, 2, 3, 4, 5, 6, 7, 8} |
| Image height | {1, 2, 3, 4, 5, 6}*min_dim_vertical |
| Image width | {1, 2, 3, 4, 5, 6}*min_dim_horizontal |
| Starting point first character | {1, 2, 3, 4}*length_single_image |
| Rotation angle | {-4, -3, -2, 0, 2, 3, 4} |
| Dilation factor | {1, 3, 5, 7} |
| Multiline | {False, True} |

Table 13: Parameters for generation of synthetic images. We used the defaults from the `PIL` (9.2.0) library for all non-reported values.

---

**Generation of synthetic images**

---

1: **function** SINGLE_LINE_IMG(encoded_word, pixel_data, labels, encoding_dictionary)
2:     Initialize an empty area for the image, based on the length of the encoded_word.
3:     **for** number in encoded_word **do**:
4:         Select randomly a row from labels matching with number.
5:         Fill the respective empty area with the EMNIST image from the pixel_data row.
6:         Add random shifts to emulate overlapping and variable distance between characters.
7:     **end for**
8:     Normalize the image and return it.
9: **end function**
10:
11: **function** WORDS2IMG(text_string, pixel_data, labels, encoding_dictionary, multiline_flag)
12:     Assign unknown characters in the text_string with a special token [UNK].
13:     Encode the text_string with the encoding_dictionary.
14:     **if** multiline_flag == False is specified **then**
15:         Call the SINGLE_LINE_IMG function with the encoded word and return the resulting image.
16:     **else**
17:         Create two encoded word lists for the two lines.
18:         Call the SINGLE_LINE_IMG function for each line and stack the resulting images vertically.
19:         Reshape the final image to the appropriate dimensions.
20:     **end if**
21:     Apply random rotation and dilation.
22:     Convert the modified image to an array and return it.
23: **end function**

---

Algorithm 1: Generation of synthetic images for data augmentation.

# G    Character Encoding with a byte-level BPE Tokenizer

| #  | Token     | Encoding | #  | Token | Encoding     |
|----|-----------|----------|----|-------|--------------|
| 1  | cls_token | [0]      | 40 | N     | [39]         |
| 2  | pad_token | [1]      | 41 | O     | [40]         |
| 3  | eos_token | [2]      | 42 | P     | [41]         |
| 4  | unk_token | [3]      | 43 | Q     | [42]         |
| 5  | "         | [4]      | 44 | R     | [43]         |
| 6  | '         | [5]      | 45 | S     | [44]         |
| 7  | (         | [6]      | 46 | T     | [45]         |
| 8  | )         | [7]      | 47 | U     | [46]         |
| 9  | *         | [8]      | 48 | V     | [47]         |
| 10 | +         | [9]      | 49 | W     | [48]         |
| 11 | ,         | [10]     | 50 | X     | [49]         |
| 12 | -         | [11]     | 51 | Y     | [50]         |
| 13 | .         | [12]     | 52 | Z     | [51]         |
| 14 | /         | [13]     | 53 | [     | [52]         |
| 15 | 1         | [14]     | 54 | ]     | [53]         |
| 16 | 2         | [15]     | 55 | ∼     | [54]         |
| 17 | 3         | [16]     | 56 | À     | [58, 63]     |
| 18 | 4         | [17]     | 57 | Á     | [58, 64]     |
| 19 | 5         | [18]     | 58 | Â     | [58, 65]     |
| 20 | 7         | [19]     | 59 | Ä     | [58, 66]     |
| 21 | 8         | [20]     | 60 | Ç     | [58, 68]     |
| 22 | 9         | [21]     | 61 | È     | [58, 69]     |
| 23 | <         | [22]     | 62 | É     | [58, 70]     |
| 24 | =         | [23]     | 63 | Ë     | [58, 71]     |
| 25 | ?         | [24]     | 64 | Í     | [58, 72]     |
| 26 | @         | [25]     | 65 | Î     | [58, 73]     |
| 27 | A         | [26]     | 66 | Ï     | [58, 74]     |
| 28 | B         | [27]     | 67 | Ñ     | [58, 75]     |
| 29 | C         | [28]     | 68 | Ò     | [58, 76]     |
| 30 | D         | [29]     | 69 | Ó     | [58, 77]     |
| 31 | E         | [30]     | 70 | Ô     | [58, 78]     |
| 32 | F         | [31]     | 71 | Ö     | [58, 79]     |
| 33 | G         | [32]     | 72 | Ú     | [58, 80]     |
| 34 | H         | [33]     | 73 | Ü     | [58, 81]     |
| 35 | I         | [34]     | 74 | Ÿ     | [59, 56]     |
| 36 | J         | [35]     | 75 | Ṅ     | [60, 57, 67] |
| 37 | K         | [36]     | 76 | …     | [61, 63, 55] |
| 38 | L         | [37]     | 77 |       | [62]         |
| 39 | M         | [38]     |    |       |              |

Table 14: Four encodings are required for classification, padding, end-of-sentence and unknown tokens. Furthermore, we observed 73 characters in our labeled material. With a byte-level BPE approach, the tokenizer required 82 encodings (integers from 0 to 81) to encode the aforementioned 77 tokens.

# H  Error analysis

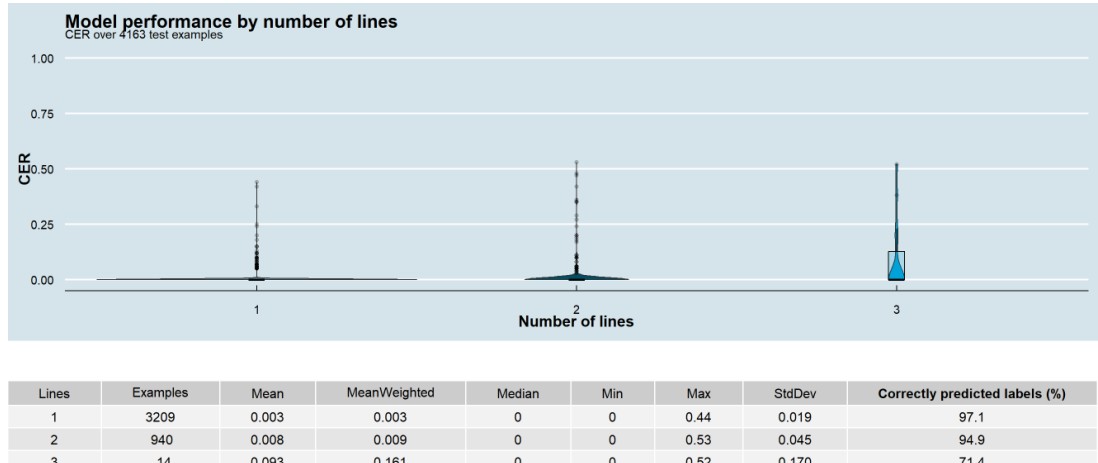

| Lines | Examples | Mean | MeanWeighted | Median | Min | Max | StdDev | Correctly predicted labels (%) |
|-------|----------|------|--------------|--------|-----|-----|--------|-------------------------------|
| 1 | 3209 | 0.003 | 0.003 | 0 | 0 | 0.44 | 0.019 | 97.1 |
| 2 | 940 | 0.008 | 0.009 | 0 | 0 | 0.53 | 0.045 | 94.9 |
| 3 | 14 | 0.093 | 0.161 | 0 | 0 | 0.52 | 0.170 | 71.4 |

Figure 6: Test set performance of our model by the number of handwritten lines.

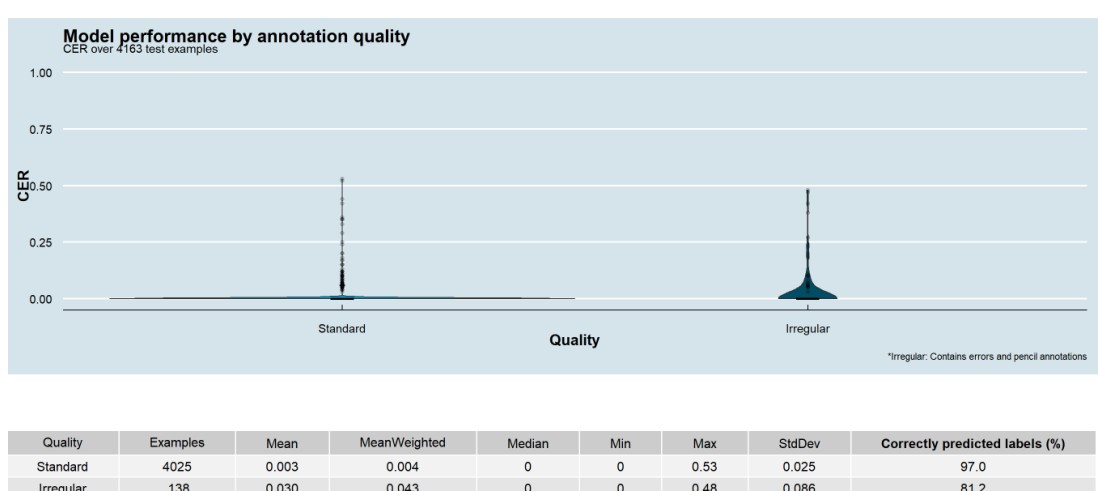

| Quality | Examples | Mean | MeanWeighted | Median | Min | Max | StdDev | Correctly predicted labels (%) |
|---------|----------|------|--------------|--------|-----|-----|--------|-------------------------------|
| Standard | 4025 | 0.003 | 0.004 | 0 | 0 | 0.53 | 0.025 | 97.0 |
| Irregular | 138 | 0.030 | 0.043 | 0 | 0 | 0.48 | 0.086 | 81.2 |

Figure 7: Test set performance of our model by annotation quality.

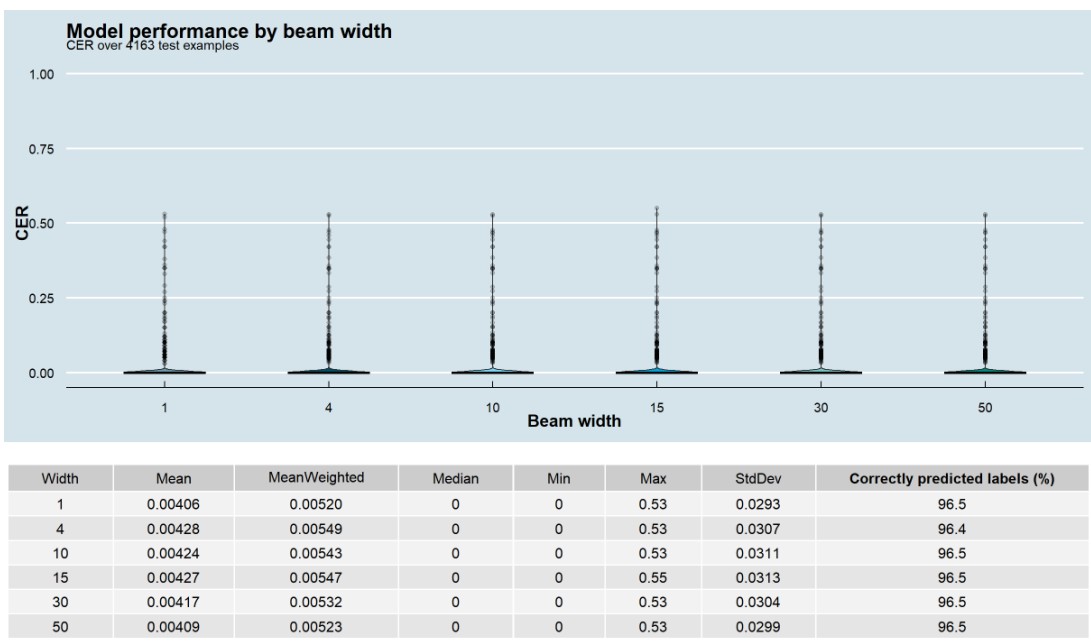

| Width | Mean | MeanWeighted | Median | Min | Max | StdDev | Correctly predicted labels (%) |
|---|---|---|---|---|---|---|---|
| 1 | 0.00406 | 0.00520 | 0 | 0 | 0.53 | 0.0293 | 96.5 |
| 4 | 0.00428 | 0.00549 | 0 | 0 | 0.53 | 0.0307 | 96.4 |
| 10 | 0.00424 | 0.00543 | 0 | 0 | 0.53 | 0.0311 | 96.5 |
| 15 | 0.00427 | 0.00547 | 0 | 0 | 0.55 | 0.0313 | 96.5 |
| 30 | 0.00417 | 0.00532 | 0 | 0 | 0.53 | 0.0304 | 96.5 |
| 50 | 0.00409 | 0.00523 | 0 | 0 | 0.53 | 0.0299 | 96.5 |

Figure 8: Test set performance by beam width.

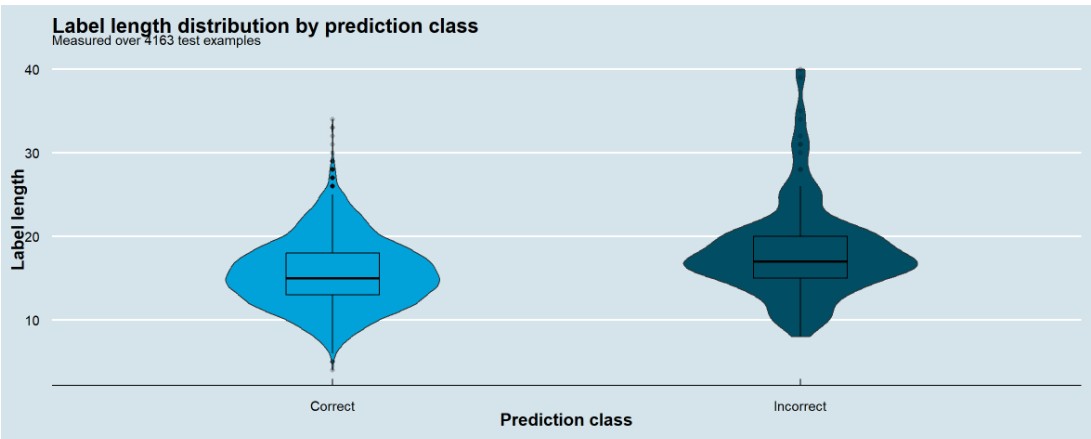

| Category | Examples | Mean | Median | Min | Max | StdDev | Correctly predicted labels (%) |
|---|---|---|---|---|---|---|---|
| Correct | 4017 | 15.662 | 15 | 4 | 34 | 4.123 | 100 |
| Incorrect | 146 | 18.267 | 17 | 8 | 40 | 5.518 | 0 |

Figure 9: Label length distribution of correct and incorrect predictions.

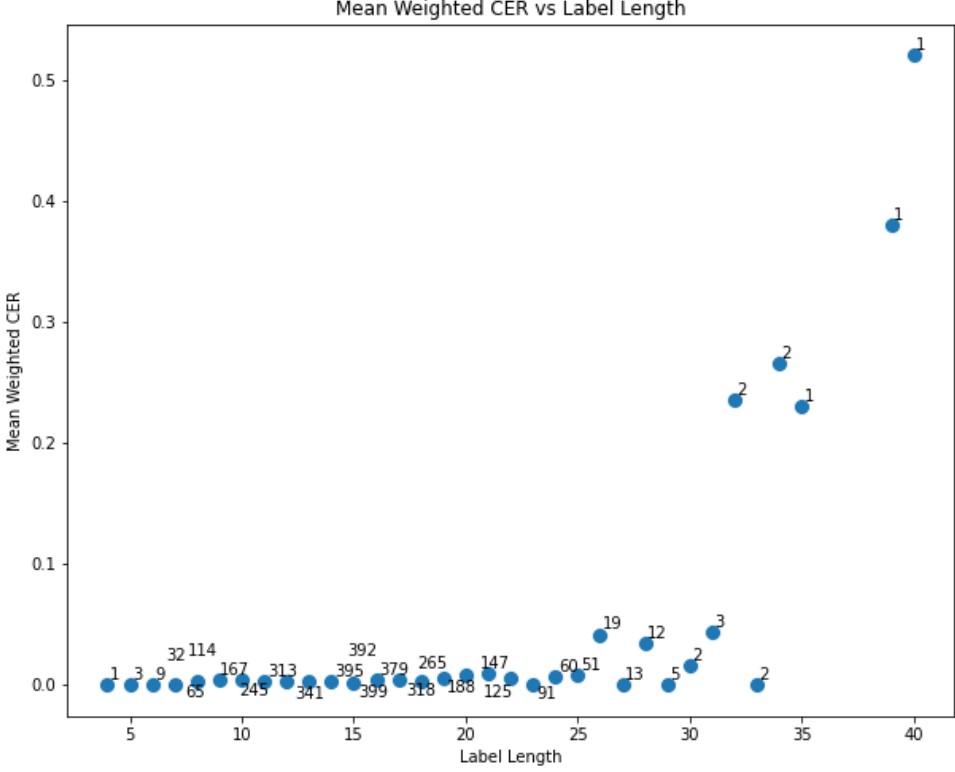

Figure 10: Mean Weighted CER by label length.

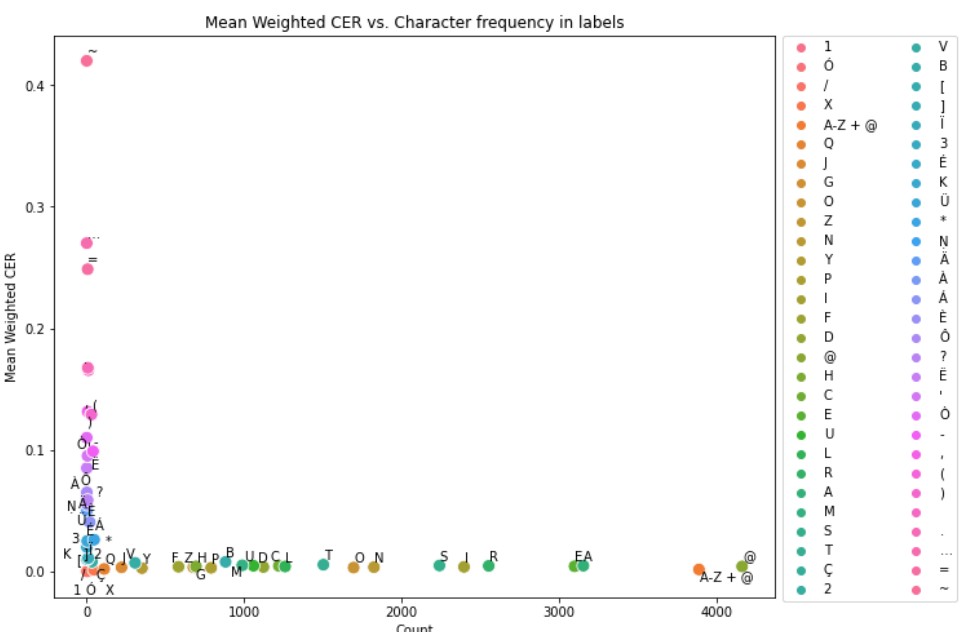

Figure 11: Mean Weighted CER by character representation in the labels.

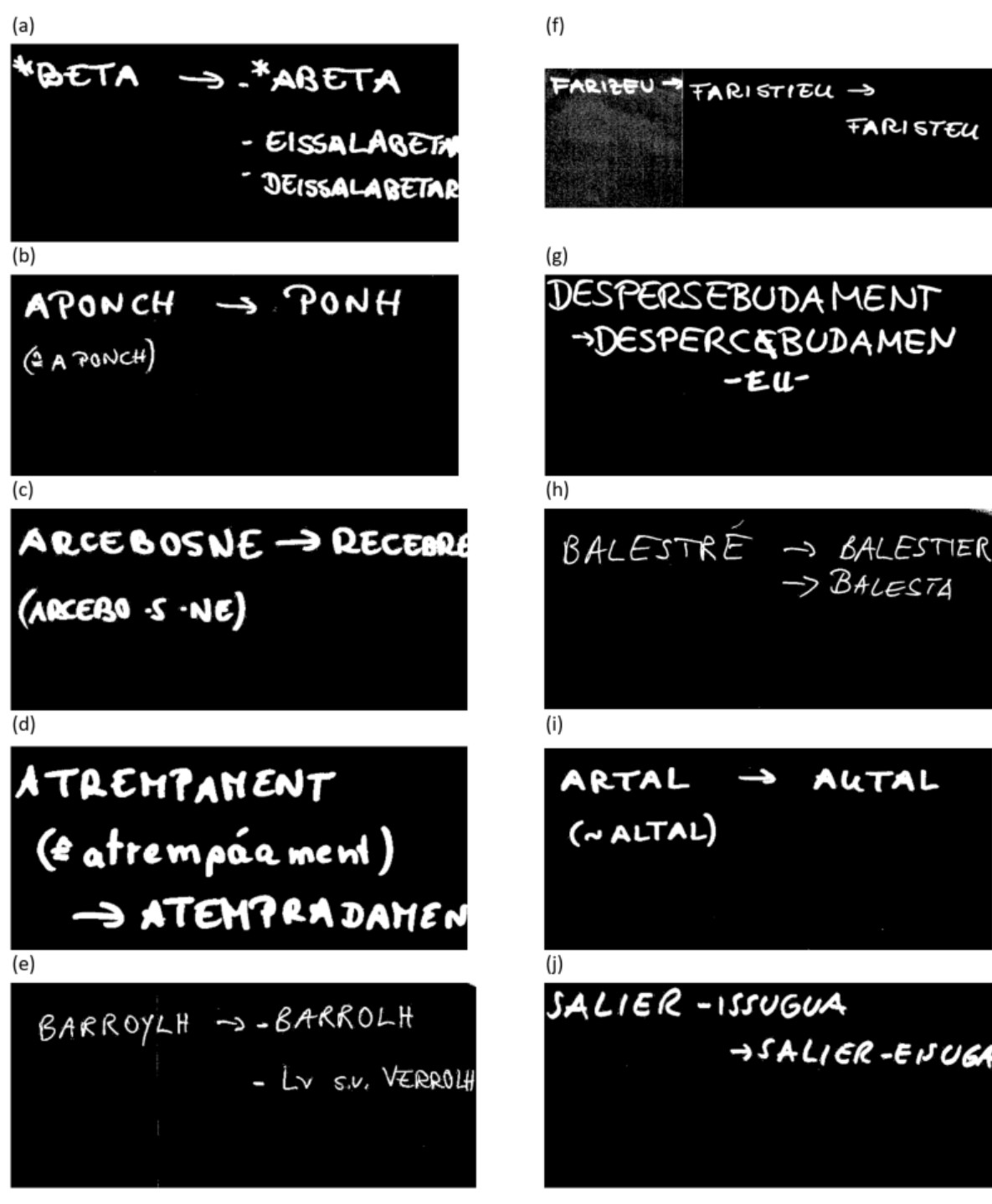

| Image | Label | Prediction | CER |
|:---:|:---:|:---:|:---:|
| (a) | *BETA@-*ABETA-EISSALABETAR-DEISSALABETAR | *BETA@*ABETA-ABETAR | 0.52 |
| (b) | APONCH(=A PONCH)@PONH | APONCH@PONH | 0.48 |
| (c) | ARCEBOSNE(ARCEBO .S .NE)@RECEBRE | ARCEBOSNE@RECEBRE | 0.47 |
| (d) | ATREMPAMENT(=ATREMPÁAMENT)@ATEMPRADAMEN | ATREMPAMENT@ATEMPRADAMEN | 0.38 |
| (e) | BARROYLH@-BARROLH -LV S.V. VERROLH | BARROILH@BARROLHER | 0.53 |
| (f) | FARIZEU@FARISTIEU@FARISTEU | FARISTIEU@FARISTIEU | 0.35 |
| (g) | DESPERSEBUDAMENT@DESPERC-EU-BUDAMEN | DESPERSEBUDAMENT@DESPERSAMEN | 0.23 |
| (h) | BALESTRE@BALESTIER@BALESTA | BALESTRÉ@BALESTIER | 0.35 |
| (i) | ARTAL( ALTAL)@AUTAL | ARTAL@AUTAL | 0.42 |
| (j) | SALIER-ISSUGUA@SALIER-EISUGA | SALIER@SALIER-ENUGA | 0.36 |

Figure 12: Worst ten predictions (based on CER) on the Old Occitan test set.

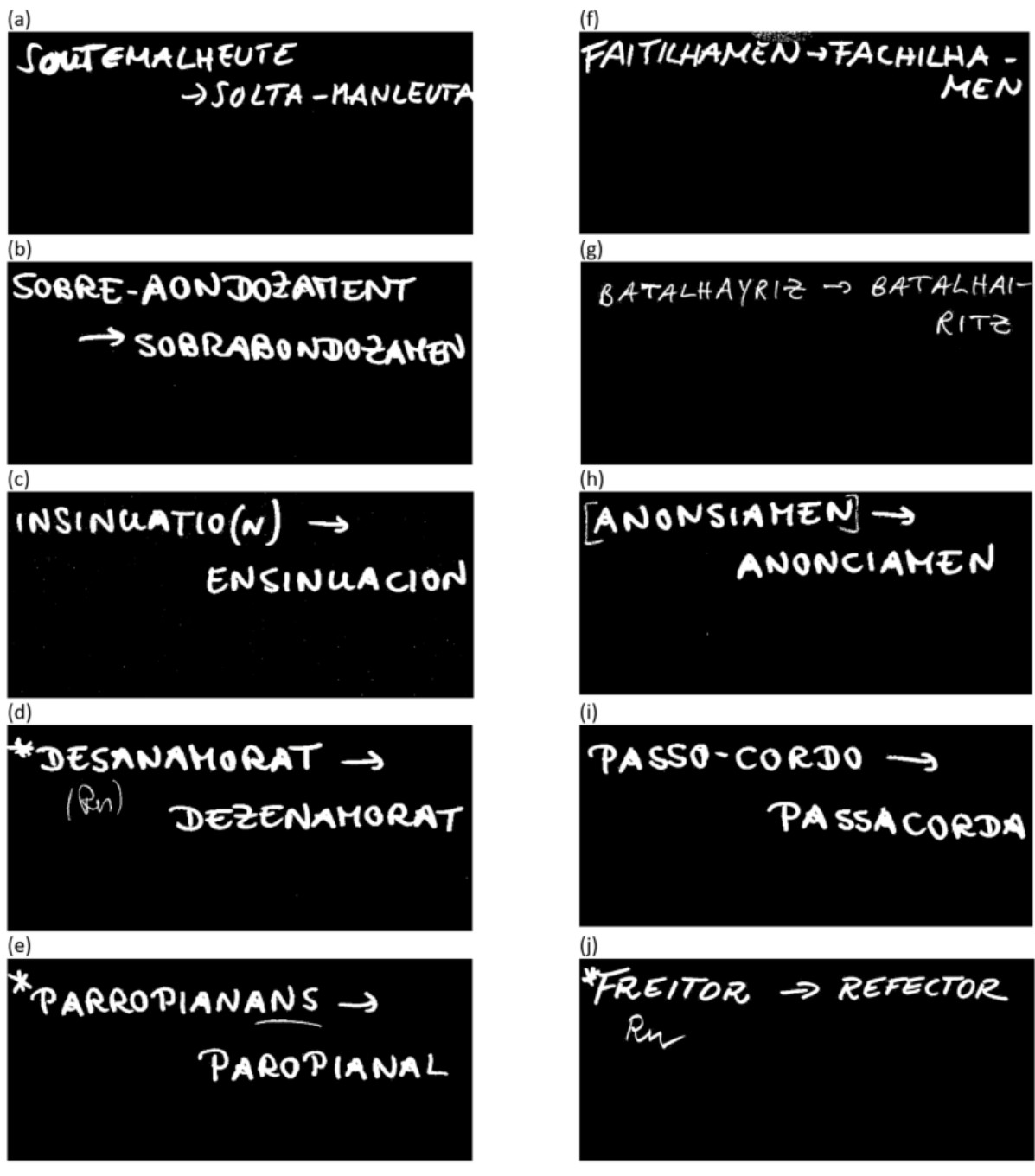

| Image | Label | Prediction | CER |
|:---:|:---:|:---:|:---:|
| (a) | SOUTEMALHEUTE@SOLTA-MANLEUTA | SOUTEMALHEUTE@SOLTA-MANLEUTA | 0 |
| (b) | SOBRE-AONDOZAMENT@SOBRABONDOZAMEN | SOBRE-AONDOZAMENT@SOBRABONDOZAMEN | 0 |
| (c) | INSINUATIO(N)@ENSINUACION | INSINUATIO(N)@ENSINUACION | 0 |
| (d) | *DESANAMORAT@DEZENAMORAT | *DESANAMORAT@DEZENAMORAT | 0 |
| (e) | *PARROPIANANS@PAROPIANAL | *PARROPIANANS@PAROPIANAL | 0 |
| (f) | FAITILHAMEN@FACHILHA-MEN | FAITILHAMEN@FACHILHA-MEN | 0 |
| (g) | BATALHAYRIZ@BATALHAI-RITZ | BATALHAYRIZ@BATALHAI-RITZ | 0 |
| (h) | [ANONSIAMEN]@ANONCIAMEN | [ANONSIAMEN]@ANONCIAMEN | 0 |
| (i) | PASSO-CORDO@PASSACORDA | PASSO-CORDO@PASSACORDA | 0 |
| (j) | *FREITOR@REFECTOR | *FREITOR@REFECTOR | 0 |

Figure 13: Best ten predictions (based on CER) of complex images from the Old Occitan test set (complexity here refers to images including multiple lines, irregular annotations, and/or rare characters, such as parentheses and hyphens).

# I Performance on a different Old Occitan data set

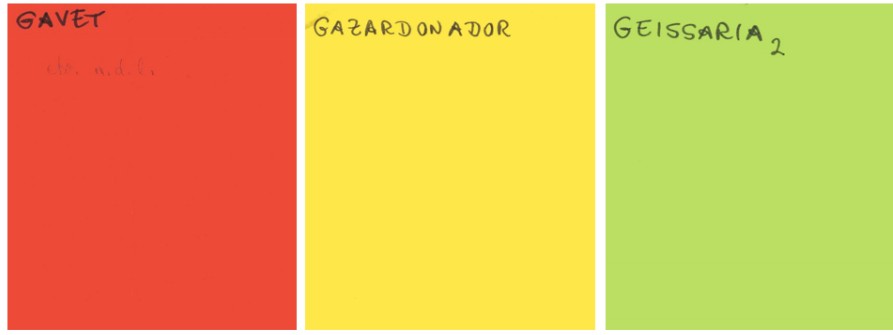

Figure 14: Examples of images from a different Old Occitan data set, comprising 316 images with simple lemmas and different background colors.

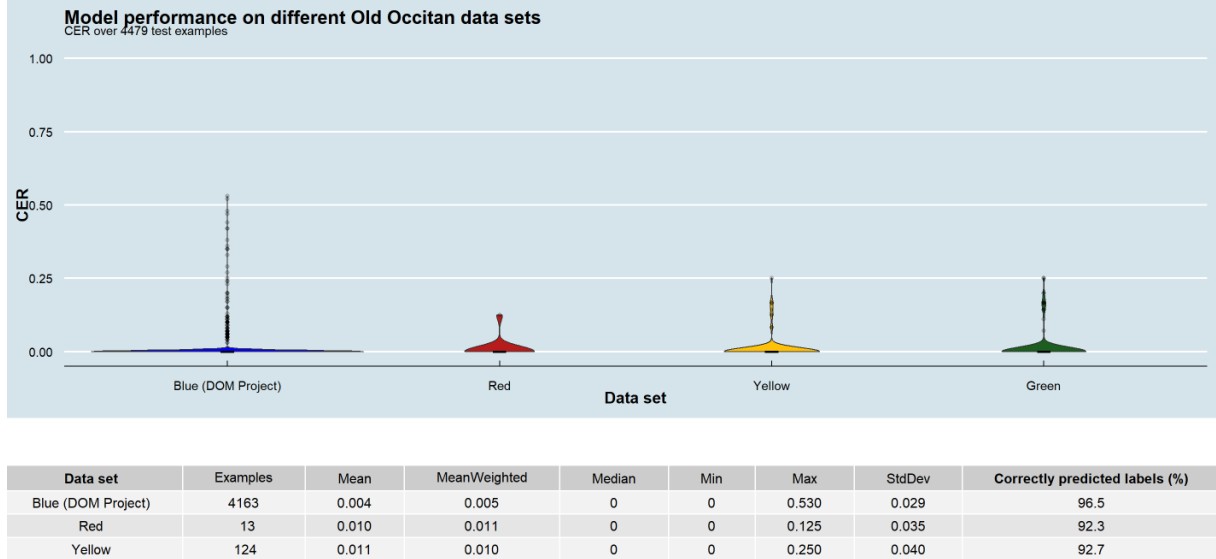

| Data set | Examples | Mean | MeanWeighted | Median | Min | Max | StdDev | Correctly predicted labels (%) |
|---|---|---|---|---|---|---|---|---|
| Blue (DOM Project) | 4163 | 0.004 | 0.005 | 0 | 0 | 0.530 | 0.029 | 96.5 |
| Red | 13 | 0.010 | 0.011 | 0 | 0 | 0.125 | 0.035 | 92.3 |
| Yellow | 124 | 0.011 | 0.010 | 0 | 0 | 0.250 | 0.040 | 92.7 |
| Green | 179 | 0.014 | 0.011 | 0 | 0 | 0.250 | 0.048 | 91.6 |

Figure 15: Test set performance comparison of our model on the Old Occitan data set (DOM Project) vs. a different Old Occitan data set, comprising 316 images with simple lemmas and different background colors. The predictions underwent an automatic post-processing step, consisting of keeping all the generated tokens until the first arrow was predicted.

## List of Acronyms

| | |
|---|---|
| **BEiT** | Bidirectional Encoder representation for Image Transformers |
| **BERT** | Bidirectional Encoder Representations from Transformers |
| **BPE** | Byte Pair Encoding |
| **CER** | Character Error Rate |
| **CNN** | Convolutional Neural Network |
| **CRNN** | Convolutional Recurrent Neural Network |
| **CTC** | Connectionist Temporal Classification |
| **DeiT** | Data-efficient image Transformer |
| **DistilBERT** | Distilled version of BERT |
| **DOM** | Dictionnaire de l'occitan médiéval (Old Occitan dictionary) |
| **GPT-2** | Generative Pre-trained Transformer 2 |
| **HTR** | Handwritten Text Recognition |
| **LSTM** | Long Short-Term Memory |
| **NLP** | Natural Language Processing |
| **OCR** | Optical Character Recognition |
| **RNN** | Recurrent Neural Network |
| **SAM** | Segment Anything Model |
| **SEEM** | Segment Everything Everywhere with Multi-modal prompts all at once |
| **SOTA** | State-of-the-Art |
| **Swin** | Shifted Window Transformer |
| **TrOCR** | Transformer OCR |
| **ViT** | Vision Transformer |
| **YOLO** | You Only Look Once |