# OpenReview forum: "Automatic Transcription of Handwritten Old Occitan Language"
_EMNLP/2023/Conference — EMNLP 2023 Main_

### Official Review · Reviewer_6eLD · 2023-07-19

**Soundness:** 4

**Excitement:**

4: Strong: This paper deepens the understanding of some phenomenon or lowers the barriers to an existing research direction.

**Missing References:**

None

**Paper Topic And Main Contributions:**

This paper presents (in great detail) how to conduct HTR for cards containing Old Occitan surface forms and the corresponding lemma(s). The authors go a long way from elaborate pre-processing methods to synthetic data generation. Moreover, the use Transformer-based architectures to train reliable models which are capable of deciphering what is written on the cards. A strong point is the carefully planned experimental design, as well as the execution of the experiments. The paper contributes novel insights and propagates the use of Swin Transformers for HTR. They compare their method to many others, thus allowing for a comparison against other systems.

**Questions For The Authors:**

Question A: What software did you use for the image processing (lines 183 to 195). PIL for everything?

Question B: You say you trained a tokenizer by setting the output tokens to 73. The decoders work on the subword levels, though. How do you justify this decision? --> by continuing reading I kind of got the answer. Ok. But still, why train your decoder from scratch and not use the pre-trained one? Have you tried employing a pre-trained decoder?

Question C: This might be provocative (but this is somehow intended): Would you agree that you push your methods quite far for a rather small dataset? Don't you think that similar results would be possible with less effort?

Question D: Would you expect better results starting from a French pre-trained LM as a decoder (like CamemBERT?)?

Question E: How big is the variation in the handwriting? Do you know how many people have participated in creating the cards?

Question F: Would you think it's worthwhile to include random warp grid distortion as a pre-processing step? C. Wigington, S. Stewart, B. Davis, B. Barrett, B. Price, and S. Cohen, ‘‘Data augmentation for recognition of handwritten words and lines using a CNN-LSTM Network,’

Question G: Did you use some kind of layout recognition?

Question H: I find parts of your synthetic data generation process counter-intuitive. The surface form and the lemma are very very close in maybe 95%(?) of the cases, i.e. there is a certain overlap. With Levenshtein you could even put a number to how close most surface forms and lemmas are. When you create your synthetic examples, they are quite off, most of the times, e.g. CARTA@AUDIR. Wouldn't you think that performance could be enhanced by creating examples which are closer to the real data?








**Reasons To Accept:**

The strong points of the paper are:
- careful experimental design and execution
- new insights with the use of Swin Transformers and training language models from scratch
- an extensive description of procedure, and further analysis provided in the appendix
- the treatment of a low-resource language and the prove that with little material, good results are attainable

As such the DH and linguistics community (esp. lexicographers, I would assume), strongly benefit from work as presented by the authors. I enjoyed the read a lot!

**Reasons To Reject:**

There are no reasons to reject the paper, but slight weaknesses:
- I find the CER graphs (violin plots, I believe?) hard to read.
- The authors mentioned this themselves, but they didn't give conventional methods the same chance as they did for Transformers. I would like to see this corrected for the final versions.
- As hinted at in Question H: the process of synthetic data generation is a bit counter-intuitive.

**Reproducibility:**

5: Could easily reproduce the results.

**Reviewer Confidence:**

5: Positive that my evaluation is correct. I read the paper very carefully and I am very familiar with related work.

**Typos Grammar Style And Presentation Improvements:**

I include general questions here. I do not expect a detailed answer for them, but are thought as food for thought for the authors. Some comments are high-level and I leave it to the authors if they want to consider them. Some are just common thoughts.

Title: Transcription ... Handwritten ... Language

009: Transformer

023: Transformer-based --> should be capitalised everywhere

049: dictionary,\footnote{} --> also for other occurrences

144: I would call this surface form instead of graphical variant

157: set instead of vocabulary

Figure 1: I struggled a bit with the term "Old Occitan cards". The cards you show contain Old Occitan cards but are not from the Odl Occitan period.

185: How do you make sure that you don't cut off text?

201: put link in a footnote

206: It would be nice to know how many lines the different corpora contain

Section 3.3: maybe provide sources which noted that this kind of data augmentation helped

---

> ### Author Rebuttal · Authors · 2023-08-25
>
> Thank you very much for the time and effort you have put into reviewing our work. Below, we answer your questions, address your concerns, and try to clear up possible misunderstandings. We will do this point by point, first copying the respective statement from your review and then providing our answer/clarification, we hope this makes reading our rebuttal for you as convenient as possible.
>
> - _Paper Topic And Main Contributions:_ This paper presents (in great detail) how to conduct HTR [...], thus allowing for a comparison against other systems.
>     - _Our Answer:_ This is a very good summary of our work, thank you very much.
>
> - _Slight Weakness 1:_ I find the CER graphs (violin plots, I believe?) hard to read.
>     - _Our Answer:_ We will try our best to improve the presentation and description of these plots in the final version of the paper. If you have any concrete suggestions about how we could present them more comprehensively, we are happy to adjust them accordingly. If your comment only concerns the degree of legibility (i.e. figure - or text - is too small?), we will definitely improve this!
>
> - _Slight Weakness 2:_ The authors mentioned this themselves, but they didn't give conventional methods the same chance as they did for Transformers. I would like to see this corrected for the final versions.
>     - _Our Answer:_ This is an interesting point that we would like to explore. Our decision was based on the promising performance of Transformer-based approaches, but we will add a comparison with a conventional method (we believe you refer to CNN+RNN-based methods?). We already looked into this a little and might include something like these options: https://link.springer.com/chapter/10.1007/978-3-031-06555-2_34, https://github.com/mittagessen/kraken or other CNN+LSTM alternatives in the final version.
>
> - _Slight Weakness 3:_ As hinted at in Question H: the process of synthetic data generation is a bit counter-intuitive.
>     - _Our Answer:_ Please see the answer to question H :-)
>
> *Questions For The Authors:*
>
> - _Question A:_ We employed several libraries for image manipulation: PIL modules for cropping, enhancement of contrast and sharpness, rotation and dilations, cv2 for enhancement of brightness, and numpy for managing images as arrays.
>
> - _Question B:_ Tokenizer: This was motivated by the lack of a known set of words and by the observed special characters linked to the specific structure of this project, such as numbers, French diacritics, arrows, punctuation marks, etc. Additionally, we considered that it would be easier for the model to predict fewer classes and provide the model with more examples per class (partially through data augmentation). For this reason, we specified the byte-level BPE to be on a character level, as illustrated in Table 14.
> Pre-trained decoder: In previous work (recognition of handwritten lemmas), we pre-trained our language decoder but the results showed a very marginal improvement (and sometimes even a negative effect on model performance). We presume that the contribution of a pre-trained decoder could be more evident in the context of handwritten documents where the context, grammar, and language structure might play a more relevant role, or when the data used during pre-training are similar to the data of the downstream task. An additional factor to be considered is that Old Occitan is a non-standardized language. This is exemplified by the existence of graphical variants/surface forms, so probably a pre-trained decoder would struggle with these (minor) deviations from the lemmas.
>
> - _Question C:_ The selection of different steps stemmed from several factors: the notable performance of Transformer-based architectures in HTR, the potential to merge diverse image encoders with language decoders (motivating the question about the best fit for our setting), and the limited availability of data (motivating different data augmentation efforts). Nevertheless, we believe that there is room for exploring numerous simplifications in forthcoming projects. For instance, we point to the study by Barrere et al. (2022) as an illustration, where they introduced a considerably smaller architecture (a 6.9 million parameters Transformer), yielding marginally inferior results compared to their baseline (a 100 million parameters Transformer).
>
> - _Question D:_ This could be interesting to explore. French and Occitan are close languages that share numerous words and characters, and a pre-trained language model such as CamemBERT could be beneficial to our project. We also consider incorporating resources from Catalan, as it shares many similarities with Old Occitan. As stated above, our tests with pre-trained language decoders have yielded mixed outcomes in the past. We presume that this observed uncertainty around the contribution of pre-trained decoders can be explained by the fact that the data used during pre-training still exerts a large influence even after fine-tuning for our (sometimes very specific, different, and data-constrained) downstream task. We intend to study this in more detail. Thank you for your suggestions!
>
> - _Question E:_ We don't have a record, but from our labeling process we have observed around 15 participants. Regarding the overall variation, we would say that over 80% of the cards follow the structure depicted in the first row of Figure 1, and the primary sources of variability identified are: stroke thickness, ink color, font size, image size, scanning quality, noise/errors, and number of lines with text.
>
> - _Question F:_ This seems to be a promising step that we could include in future work, thank you! During data augmentation, we tried to generate word pairs that were not necessarily present in the original data to increase the variability of the training examples. However, applying a random warp grid distortion step to both existing and new (synthetic) word pairs could be definitely beneficial. The paper also highlights good results of RWGD across multi-author datasets of different languages, centuries, and various degrees of legibility, so we will consider this approach in future projects.
>
> - _Question G:_ We didn't include this step in our method due to the rather simple and clean structure of our data, but will probably need it in future projects, where the handwritten text requires recognition and segmentation. Layout recognition could be also a beneficial step in this project in order to accurately identify the relevant parts of the images (in particular for the more challenging cards (with multiple lines of text, including annotation errors), where the model struggles to correctly distinguish the graphical variants/surface forms from the lemmas).
>
> - _Question H:_ This was indeed our first approach, we started generating word pairs for words that shared a maximum Levenshtein distance of 2 (as this was the observed median distance in the training material). However, we then moved to a more flexible approach to increase the number (and variability) of word pairs. Generating word pairs based on the Levenshtein distance might generate word pairs that are closer to the original material, but it also represents an additional constraint that reduces the number of available examples per character, which is critical to their accurate prediction, as visualized in Figure 11 (Appendix).
> As we didn't observe any performance decrease, we kept this more flexible strategy. Maybe this flexibility can lead to better generalization on slightly different data (but still of a similar structure) while not harming performance for our use case.
>
> - _Typos Grammar Style And Presentation Improvements:_ Thank you very much again for such detailed and helpful comments, we will work through all of them and adjust our paper accordingly.

---

### Official Review · Reviewer_udXX · 2023-08-06

**Soundness:** 4

**Excitement:**

4: Strong: This paper deepens the understanding of some phenomenon or lowers the barriers to an existing research direction.

**Paper Topic And Main Contributions:**

Old Occitan is a language spoken in 11th-16th centuries in France, Spain and Italy. Authors propose a method for OCR of this language from handwritten text and show that their approach beats the state of the art systems.
Data collection and augmentation details are well described although it is not clear if it is made public.
Algorithm uses a image encoder followed by a BERT-based decoder. The algorithm is also compared with GPT-2.

**Questions For The Authors:**

Is the dataset being made public?

**Reasons To Accept:**


The paper is well motivated and well written. The dataset is described well. Results from the approach is promising.

**Reasons To Reject:**


The scope of the handwritten text (capital handwritten letters, two words per card) is quite limited. It will be good to extend it to handwritten documents.


**Reproducibility:**

3: Could reproduce the results with some difficulty. The settings of parameters are underspecified or subjectively determined; the training/evaluation data are not widely available.

**Reviewer Confidence:**

5: Positive that my evaluation is correct. I read the paper very carefully and I am very familiar with related work.

---

> ### Author Rebuttal · Authors · 2023-08-25
>
> Thank you very much for the time and the effort you have put into reviewing our work. Below, we answer your questions, address your concerns and try to clear up possible misunderstandings. We will do this point by point, first copying the respective statement from your review and then providing our answer/clarification, we hope this makes reading our rebuttal for you as convenient as possible.
>
> - _Paper Topic And Main Contributions:_ Old Occitan is a language spoken in [...] uses a image encoder followed by a BERT-based decoder. The algorithm is also compared with GPT-2.
>     - _Our Answer:_ This is a very good summary of our work, thank you very much. Please find our comments about reproducibility below :)
>
> - _Reasons To Reject:_ The scope [...] quite limited. It will be good to extend it to handwritten documents.
>     - _Our Answer:_ This is a valid point. However, we would like to mention that the model has shown good performance (after automatic postprocessing) on a further Old Occitan dataset that only contains single words (Figure 14, in the Appendix), demonstrating its effectiveness under a slight domain shift (Figure 15, in the Appendix). In future work, we plan to explore the potential of our approach to handwritten documents of Old Occitan and other languages, and our aim is to effectively expand its capabilities to a wider range. We hope that this work is an additional step for this under-resourced language, paving the way for further advancements.
>
> - _Questions For The Authors:_ Is the dataset being made public?
>     - _Our Answer:_ Yes, our model, datasets, and code are already available in the anonymous GitHub repository (link in the Abstract, line 28), and will be publicly available on Hugging Face upon publication. The appendix also provides a comprehensive list of hyperparameters and settings and all the scripts are available and ready to use.
>
> - _Reproducibility: 3_
>     - _Our Answer:_ As stated before, all of our data, models, and code are already available in an anonymous GitHub repository and will be made publicly available on GitHub and Hugging Face. We hope to have cleared the uncertainties on that end and we would be very grateful if you could take these clarifications into account when reassessing the degree of reproducibility of our work.

---

### Official Review · Reviewer_GHMV · 2023-08-07

**Soundness:** 3

**Excitement:**

3: Ambivalent: It has merits (e.g., it reports state-of-the-art results, the idea is nice), but there are key weaknesses (e.g., it describes incremental work), and it can significantly benefit from another round of revision. However, I won't object to accepting it if my co-reviewers champion it.

**Paper Topic And Main Contributions:**

This paper presents a data augmentation technique for the Old Occitan language and benchmarks a transformer model trained with this data against some SOTA HTR models. These transformers employ and test a combination 4 vision encoders and 2 textual encoders.

**Questions For The Authors:**

Is the fine-tune  TrOCR model trained on same data as other  (Swin + BERT) model, if not then what's the reasons  and how is this comparison fair ?

**Reasons To Accept:**

The paper encompasses an exhaustive benchmarking of the performance of  HTR recognition for Old Occitan language using smart data augmentation to deal with scarcity. The final model also shows significance gains from the augmentation as compared with the real data itself helping the model trained with this data to outperform open-source solutions.

**Reasons To Reject:**

Hard to identify the core novelty proposed in the technique:
- it seems, that smart data augmentation for this language is a core contribution. However, all the proposed image transformations are well-known and widely used. Any novel augmentation specific (to this language or any HTR dataset) needs to be clearly stated and compared. Since the paper doesn't compare with any other augmentation (generic to vision or HTR) method, hard to contextualize the contribution.

- In terms of the model all 4 vision encoders and 2 text encoders are used off the shelve, and no new architectural changes are proposed (including the fusion of encoders). Though there are some other native HTR transformer-based models mentioned by authors like Barrere et al. (2022), the authors don't compare with those.

- Lastly seems like fine-tuned TrOCR which is the closest transformer baseline compared with is not fine-tuned on the target domain dataset (as from line 433-443, author explain domain mismatch), in such case lower performance is expected. Why doesn't the author fine-tune  TrOCR on the same data and test against the best (Swin + BERT) model

**Reproducibility:**

3: Could reproduce the results with some difficulty. The settings of parameters are underspecified or subjectively determined; the training/evaluation data are not widely available.

**Reviewer Confidence:**

1: Not my area, or paper was hard for me to understand. My evaluation is just an educated guess.

---

> ### Author Rebuttal · Authors · 2023-08-25
>
> Thank you very much for the time and effort you have put into reviewing our work. Below, we answer your questions, address your concerns, and try to clear up possible misunderstandings. We will do this point by point, first copying the respective statement from your review and then providing our answer/clarification, we hope this makes reading our rebuttal for you as convenient as possible.
>
> - _Paper Topic And Main Contributions:_ This paper presents a data augmentation technique [...] 4 vision encoders and 2 textual encoders.
>     - _Our Answer:_ Data augmentation is indeed a fundamental step, but not the focus of the paper. Our core contribution is that we evaluate eight combinations of encoder-decoder architectures as a first step, and provide insights through our experiments into the benefits of a combination of a Swin image encoder with a BERT-based language decoder (trained from scratch) in the context of a low-resource language. In a second, auxiliary step, we enhance its performance through different data augmentation techniques and provide insights about its performance compared to well-established open-source and commercial alternatives. Finally, we hope to contribute to the HTR community by making our model, datasets, and code publicly available on GitHub/Hugging Face.
>
> - _Reasons To Accept:_ The final model [...] outperform open-source solutions.
>     - _Our Answer:_ One important further point that we would like to add is that we also outperform a well-established closed-source solution (Google Cloud Vision, see Table 2 and Figure 4) that has proven its effectiveness in the context of low-resource languages.
>
> - _Reasons To Reject 1:_ Hard to identify the core novelty proposed in the technique:
>     - _Our Answer:_ It is correct that we do not introduce a novel architecture. Instead, the novelty of our works lies in the insights gained through exhaustive experiments of model combinations (Table 3) and benchmarking (Figure 4) in the specific context of this particularly low-resource language. In our ablation study, we also measure the contribution of straightforward vs. more elaborate data augmentation techniques to the performance of our best model (Table 4). Additionally, we believe that the novel resources (curated data, high-performing models, and code) we make publicly available will be a contribution to the community (in particular Digital Humanities and Linguistics).
>
> - _Reasons To Reject 2:_ it seems, that smart data augmentation for this language is a core contribution. However, all the proposed image transformations are well-known [...] hard to contextualize the contribution.
>     - Our Answer: These data augmentation techniques play a valuable role in addressing data scarcity and enhancing our models' performance, though they are not the central focus of our work and therefore we did not include a comprehensive benchmarking of this particular step. Nevertheless, the ablation study (Table 4) provides a simple comparison of straightforward techniques such as dilation and rotation against the more elaborate generation of EMNIST-based synthetic images. This contrast highlights the notable advantages of the latter, particularly as the volume of synthetic images increases.
> Still, we are aligned in the view that delving into and evaluating alternative data augmentation techniques (like GANs or Random Warp Grid Distortion -as suggested by Wigington et al.), is an important step to gauge their impact on model performance and we will consider this in our future research.
>
> - _Reasons To Reject 3:_ In terms of the model all 4 vision encoders and 2 text encoders are used off the shelve, and no new architectural changes are proposed [...] Though there are some other native HTR transformer-based models mentioned by authors like Barrere et al. (2022), the authors don't compare with those.
>     - _Our Answer:_ In addition to our response to "Reason to Reject 1", we would like to add that one of our main objectives is to compare the models (as they were published, with their base configuration) and highlight the best mix (with the help of data augmentation) in the context of a low-resource language. As part of our future research, we intend to explore and propose more efficient architectures, including an optimized fusion of the encoder and decoder as well as customized loss functions to balance the information coming from both images and text.
> Regarding the "Light transformer" model introduced by Barrere et al. (2022): To the best of our knowledge, the model isn't accessible to the public, thus preventing its inclusion in our comparative analyses. Nevertheless, we have contacted the authors and we plan to include it in the final version upon a positive response from the authors.
>
> - _Reasons To Reject 4:_ Lastly seems like fine-tuned TrOCR [...] is not fine-tuned [...], in such case lower performance is expected. Why doesn't the author fine-tune TrOCR on the same data and test against the best (Swin + BERT) model
>     - _Our Answer:_ This is a very crucial point, and we would like to clarify that we DO FINE-TUNE the TrOCR on the same data as our best-performing model to ensure a fair comparison. The results are reported in the Benchmarking section, along with other architectures (Figure 4).
> In lines 433 – 443, we discuss the performance of the fine-tuned TrOCR and speculate on the reasons for the performance differences (e.g. we presume that the effect of pre-training with millions of images of English sentences could be related to the generation of rather long predictions - even after fine-tuning with Old Occitan data). We will make sure that this is conveyed with more clarity in the final version of the paper.
>
> - _Questions For The Authors:_ Is the fine-tune TrOCR model trained on same data as other (Swin + BERT) model, if not then what's the reasons and how is this comparison fair?
>     - _Our Answer:_ As stated above, we have indeed fine-tuned the TrOCR using the same dataset as our most successful model. The outcomes of this fine-tuning process have been detailed in the Benchmarking section and visually presented in Figure 4. As a side comment: The performance of TrOCR (without finetuning) in Old Occitan (or in many other languages different from English) is remarkably low, the predictions are usually random English words that have almost no connection to the input handwritten text. This can be verified under the following link: https://huggingface.co/spaces/nielsr/TrOCR-handwritten
>
> - _Reproducibility: 3_
>     - _Our Answer:_  As stated in the paper, our model, datasets, and code are already available in the anonymous GitHub repository (link in the Abstract, line 28), and will be publicly available on Hugging Face upon publication. The appendix also provides a comprehensive list of hyperparameters and settings and all the scripts are available and ready to use. We hope to have cleared the uncertainties on that end and would be grateful if you could take these clarifications into account when reassessing the level of reproducibility of our work.
>
> - _Soundness: 2_
>     - _Our Answer:_ We hope to have answered all the questions to your satisfaction and also to have clarified your concerns about the fairness in our experiments and benchmarking, in particular with regards to the fine-tuning of TrOCR with the same data as our best-performing model. We would greatly appreciate it if you could consider our clarifications while reassessing the soundness of our work. We hold the belief that a rating of 2 (“...major technical/methodological problems”) may have potentially been influenced by the ambiguities surrounding the fairness of our methodology.

---

### Meta-Review · Area_Chair_PM14 · 2023-09-20

**Recommendation:** 5

**Metareview:**

The paper is well motivated and well written. The dataset is described well. The authors have considered an exhaustive set of approaches in the experimentation setup and have conducted a detailed analysis. There are new and possibly reusable insights with the use of Swin Transformers and training language models from scratch.

Some points to be improved:
- Add conventional methods like CNN+LSTM in the experimental analsysis
- Discuss negative results as well on  the synthetic data generation process
- Hard to identify the core novelty proposed in the technique:

---

### Decision · Program_Chairs · 2023-10-07

**Decision:**

Accept-Main

**Comment:**

The paper is well motivated and well written. The dataset is described well. The authors have considered an exhaustive set of approaches in the experimentation setup and have conducted a detailed analysis. There are new and possibly reusable insights with the use of Swin Transformers and training language models from scratch.

Some points to be improved:
- Add conventional methods like CNN+LSTM in the experimental analsysis
- Discuss negative results as well on  the synthetic data generation process
- Hard to identify the core novelty proposed in the technique: